



# Summer valley-floor snowfall in Taylor Valley, Antarctica from 1995 - 2017

Madeline E. Myers[1,2], Peter T. Doran[1], Krista F. Myers[1]

[1]Department of Geology and Geophysics, Louisiana State University, Baton Rouge, LA, USA

[2]Department of Geography and Planning, Queen's University, Kingston, Ontario, Canada

*Correspondence to*: Madeline E. Myers (madeline.myers@queensu.ca)

**Abstract.** In polar, coastal areas like Taylor Valley, snowfall is predicted to increase under warming conditions as reduced sea ice increases open water area and evaporation potential, thereby creating conditions that would facilitate precipitation. Taylor Valley is a mosaic of glaciers, valley-bottom ice-covered lakes, ephemeral streams and dark, rocky soils. Ecosystems are both

light- and nutrient-limited and rely on seasonally available liquid water. Although Taylor Valley receives minimal snowfall annually, light snow cover during summer months reduces radiation for primary productivity and slows melting by increasing the local albedo. Snowfall has been measured at four sites in Taylor Valley since 1995. Daily photographs at the Lake Hoare station in the central portion of the valley record snow cover since 2007 and augment the automated precipitation measurements. Here, we focus on valley-floor snowfall due to its effect on ecosystems in the valley-bottom lakes and streams.

Precipitation increased by 3 mm water equivalent (w.e.) a$^{-1}$ from 1995 to 2009, then decreased by 1 mm w.e. a$^{-1}$ through 2017. Since 2009, annual snowfall in Taylor Valley ranges from 1 to 30 mm w.e. High snowfall during the Spring near the coast is indicative of high summer snowfall at the more inland Lake Bonney station (r$^2$ = 0.66; p < 0.05). In contrast, the average fraction of days with snow on the ground tripled at Lake Hoare after 2011, primarily during Spring and Fall. Fall snow persistence at Lake Hoare has been increasing by ~1 day per year since the start of the record in 2007, although Spring snow

cover exhibits no trend. In agreement with previous studies, regression analysis revealed no correlation of snow cover or snowfall with sea ice extent or mean temperatures. Strong seasonality and interannual variability underscores the complexity of precipitation and snow persistence controls in Taylor Valley. In regions where snow cover contributes more to the radiation budget than the hydrologic budget, photographs are the most reliable method for monitoring precipitation. The results of this study highlight the importance of continued monitoring of precipitation throughout Taylor Valley. The establishment of coastal

and inland snow cover monitoring stations would augment point observations of snow cover and add spatial complexity to our present understanding of the expected hydrologic and ecosystem response to climate change in Taylor Valley.



# 1 Introduction

The McMurdo Dry Valleys (MDVs) are located along the coast of Antarctica in southern Victoria Land  (Figure 1). Taylor Valley (TV) lies within the MDVs. The Transantarctic Mountains buffer TV from the East Antarctic Ice Sheet and allow them to remain ice-free with the exception of glaciers that flow from the surrounding mountains (Chinn, 1990). Ephemeral streams supply glacial melt to closed-basin lakes along the valley-bottom. Water input from the glaciers is balanced by evaporation and sublimation of the permanent lake ice (Doran et al., 1994; Dugan et al., 2013) and perennially melted perimeter (Chinn, 1993). Seasonal melting of ground ice and snowpacks provide moisture to the soil and transport a nearly equivalent amount of nutrients to the lakes as the streams (Levy et al., 2011). TV is bounded by the Ross Sea to the east and Taylor Glacier to the west (Figure 1). Average annual valley bottom air temperature is 18.5°C (Obryk et al., 2020) and previous work (Fountain et al., 2010) found TV only receives 3 to 50 mm snow water equivalent (SWE) annually with greater accumulation nearest the coast.

The McMurdo Long Term Ecological Research (MCM LTER) project has supported research in Taylor Valley since 1992. TV is a polar desert bounded to the east by the Ross Sea and to the west by the Transantarctic mountains, which allow TV to remain ice-free (Denton and Marchant, 2000). Microbial productivity peaks during the summer months when light availability drives photosynthesis in the lakes (Fritsen and Priscu, 1999) and melt from the surrounding alpine glaciers and snowpacks initiates photosynthesis in the streams (McKnight et al., 1999) and soils (Gooseff et al., 2003; Levy et al., 2011). TV ecosystems are highly sensitive to small climatic fluctuations due to the limited availability of liquid water and energy (Moorhead and Priscu 1998).

Snow cover regulates primary productivity and hydrologic connectivity in TV. It blocks radiation that would be used for photosynthesis by underlying organisms (Bolsenga et al., 1996; Hawes 1985).  It increases the albedo of the dark, rocky landscape, including icy surfaces (Bergstrom et al., 2020), and thereby reduces melt generation during summer months (Perovich and Polashenski, 2012). Additionally, summer snow cover functions as a form of surface humidity and slows melting of subsurface ice (Kowalewski et al., 2006; Hagedorn et al., 2007; McKay 2009) and recurring snowpacks, both of which are an important moisture source for soil ecosystems during the melt season (Gooseff et al., 2013; Zeglin et al., 2009).

Under warming conditions, snowfall is expected to increase in polar coastal regions due to increased evaporation potential of nearby water bodies under reduced sea ice conditions (Bintanja and Selten, 2014). In TV, reduced sea ice concentrations in the nearby Ross Sea may increase snowfall and snow cover, however only under certain large-scale meteorological conditions. Snowfall is controlled by both moisture availability and prevailing wind patterns (Patterson et al., 2005). When the Amundsen Sea Low (ASL), a low pressure center, is positioned north of the Ross Ice Shelf, air masses are carried across the Ross Sea (Patterson et al., 2005). In this case, sea ice extent in the Ross Sea may impact snowfall in TV. In other cases, the ASL is positioned in the Amundsen Sea and air masses are transported across the West Antarctic Ice Sheet (WAIS) (Patterson et al., 2005) and precipitation in Taylor Valley may not be affected by moisture availability in the Ross Sea. Bertler et al. (2006) show El Niño Southern Oscillation (ENSO)-Antarctic teleconnections can influence the position of





the ASL. During La Niña, the ASL encourages moisture transport across the Ross Sea and El Niño encourages the opposite. Turner (2004) discusses factors affecting the strength of tropic-Antarctic teleconnections in detail, and they will not be repeated here; however, it is important to note that the teleconnection strength varies on monthly to decadal timescales. Fogt and Bromwich (2006) show that the teleconnection is strongest in Spring and Summer and insignificant in Fall and Winter. Snow

should be analysed on similar time scales in order to determine a true climate signal and anticipate associated changes.

Upon deposition, snow is redistributed by wind and is lost to the atmosphere by sublimation and evaporation and can persist for weeks (Eveland et al., 2013). Although ablation is dominated by sublimation and evaporation (Gooseff et al., 2011; Harpold and Brooks, 2018), snow-patches may melt and percolate into the soil. Gooseff et al. (2003) observed subnivean soil moisture was 9% compared to exposed soils at 0.5 %. Nearest the coast, the Lake Fryxell (FRLM) basin experiences cooler

temperatures, higher relative humidity (RH) and lower wind speeds than the 20 km inland Lake Bonney (Doran et al., 2002a). The RH and wind speed gradient drive higher sublimation rates inland (Doran et al., 2002a). Windy conditions can redistribute snow from the surrounding mountain peaks to the valley floor and appear as snowfall events (Fountain et al., 2010).

Fountain et al. (2010) discussed snow in TV from 1995 through 2006 from automated measurements by ultrasonic distance transducers, weighing bucket gauges, and antifreeze tipping buckets. This record covered the climate cooling period

identified by Doran et al. (2002b). Recently, Obryk et al. (2020) revealed a shift in temperature trend over the last three decades starting in 2005 from cooling to no trend. Precipitation in Taylor Valley has not been re-evaluated since the observed temperature shift and this is the first opportunity to test whether we see a similar shift in annual snowfall. Additionally, we will address seasonal-scale trends in snowfall and snow persistence to highlight variability that may be dependent on shorter-lived factors such as atmospheric oscillations. Finally, we reveal shortfalls of presently established methods for monitoring

precipitation in TV and highlight how daily photographs improve them.

## 2 Methods

### 2.1 Automated Weather Stations (AWS)

Snowfall has been recorded at 4 meteorological stations in TV run by Campbell Scientific CR10X dataloggers. Accumulation was measured by sonic rangers (Campbell Scientific SR50), weighing bucket gauges (Belfort), and an antifreeze

tipping bucket (Texas Electronics TE525MM) since 1994 (Doran and Fountain, 2019a; Doran and Fountain, 2019b; Doran and Fountain, 2019c; Fountain and Doran, 2019). Weighing bucket gauges and tipping buckets employ a Nipher wind shield. For windspeeds less than 5 m s$^{-1}$, the Nipher wind shield captures 95 to 111% of precipitation (Goodison, 1978). We processed data in accordance with the methods established by Fountain et al. (2010), who calculated precipitation from the difference in accumulation between adjacent days when it exceeded 0.5 mm water equivalent (w.e.). They excluded precipitation events

measured when daily average wind speeds exceed 5 m s$^{-1}$ which could convey snow from the surrounding peaks to the valley-floor. More details on station set up and data processing are described in detail by Fountain et al. (2010). Data are accessible



from the MCM LTER website (http://mcm.lternet.edu/). Snow depth measured from ultrasonic distance rangers were converted to mm w.e. with a snow density of 83 kg m$^{-3}$ measured near Lake Fryxell in late December of 2018. Volume was derived from measurements of freshly fallen snow in a 0.5 m x 0.5 m clean, smooth surface. The snow was melted in a sealed bag and weighed to calculate density.

Similar to a water year in hydrological analyses, a "snow year" was defined as being from May 1 to April 30 of the following year, which roughly coincides with the final sunset of the season. Total measured precipitation was calculated for each snow year as the sum of differences in daily averages that exceed the 0.5 mm w.e. threshold defined above. The use of a snow year allows better comparison to processes governed by light availability and melt generation. Here, we focus on Spring, Summer, and Fall with Winter excluded for the same reason. Spring begins with first light and ends November 15. Fall begins

February 15 and ends with the final sunset. Dates coincide with statistically distinct climate conditions (Obryk et al., 2020). Accumulation was calculated for each season and snow year with over 75% data available. To test the hypothesis of local sea ice influencing snow in TV, we conducted regression analysis of snow volume at FRLM by season with distance to sea ice edge from Kim et al. (2017). Where more than 10-years continuous data were available, we applied nonparametric Pettitt test statistics to determine pivot points (Pettitt, 1979). If three or fewer consecutive years were missing, data were linearly

interpolated. The Pettitt test statistic is commonly used to identify changepoints in continuous climate and hydrologic timeseries (Kundzewicz and Robson, 2000). Where changepoints were detected, nonparametric Mann-Kendall test was run before and after pivot points to determine if there was a trend. The trend was then characterized by Sen's slope, which is the median slope through pairs of points and is insensitive to outliers (Sen, 1968).

### 2.2 Snow Persistence

Terrestrial photos have proven to be a valuable data source for monitoring snow cover (Härer et al., 2018; Salzano et al., 2019; Portenier et al., 2020). The meteorological snow record has been evaluated against Campbell Scientific CC640 camera photos at Lake Hoare taken every 6 hours since October of 2007 (Figure 2). Total number of days with snow on bare ground was determined from visual inspection of the camera record. Any new accumulation of snow was considered an event. Persistence for each event was calculated by subtracting the end date from the initial date with snow on the ground for that

event. Images when the camera was covered by snow were excluded as missing data. Data were broken into seasons and evaluated similar to the meteorological record.

All statistical analyses were conducted in SPSS Statistics (version 24) unless specified otherwise. Summer snow cover at Lake Hoare was reconstructed for the years prior to the establishment of the camera from glacier accumulation data. Accumulation measurements have been taken since 1993 for Commonwealth Glacier (Gooseff and Fountain, 2019). This study

utilizes data from stake 23 in the accumulation zone. Accumulation is typically measured between November and January each year. Here, we use the percentage of days with snow on the ground derived from the camera between summer accumulation measurements on Commonwealth Glacier. The relationship between accumulation and the percentage of snow





cover days for the summer at Lake Hoare is derived from simple linear regression of accumulation and the percentage of days with snow cover between those two accumulation measurements since 1993. Regression analysis was conducted on snow cover and sea ice extent similar to the analysis done for FRLM snowfall.

## 3 Results

### 3.1 Precipitation Volume

Spatial and temporal patterns of precipitation in TV were determined from sonic distance rangers and weighing bucket gauges. First, we discuss annual precipitation across all stations (Figure 3). Again, we focus on snow between first and last light in August and May and will refer to it as a "light season" henceforth so as not to be confused with annual or Summer snowfall. Light season precipitation at all stations typically trend similarly but at different magnitudes. The inland Lake Bonney AWS (BOYM) and coastal FRLM AWS correlate where data overlap ($r^2 = 0.66$; $p < 0.05$). Over the course of the record, snowfall at both BOYM and FRLM reach maximums in 2007 although Pettitt change point test does not identify any significant changepoints in either dataset. The Mann-Kendall test identifies significant increasing precipitation (4.5 mm w.e. $a^{-1}$; $p < 0.05$) at FRLM prior to 2007 and decreasing (3.5 mm w.e. $a^{-1}$; $p < 0.01$) from 2007 to 2013. Where data are available, the highest mean volume of precipitation is at FRLM which receives an average of 11.5 mm w.e. from August through May, closely followed by the more coastal EXEM which received 10 mm w.e. The Lake Hoare (HOEM) ultrasonic ranger records the lowest snowfall (3.5 mm w.e.), but the weighing bucket gauges at HOEM and BOYM both record 5.5 mm w.e. From August through May, Taylor Valley receives an average of 7.5 mm w.e ranging from 1 to 25.5 mm w.e.

We assess the seasonal variability of precipitation at each station because of seasonal differences in atmospheric influences on precipitation (Figure 4). For all stations, the lowest accumulation occurs in the Summer and is fairly consistent at *c.* 0.5 mm w.e. at all stations. During Spring and Fall, however, there appears to be more spatial control on snowfall where EXEM and FRLM receive similar volumes and HOEM and BOYM receive similar lower volumes. EXEM and FRLM receive 4.5 and 5.5 mm w.e. respectively in Spring and 4.0 and 4.5 respectively during the Fall. HOEM and BOYM receive 1.5 and 2.0 mm w.e. respectively in Spring and 1.0 and 2.0 mm w.e. in Fall. Consequently, over a third of August through May snowfall w.e. in TV occurs in the Lake Fryxell basin (FRLM and EXEM) during the Spring and another third in the Fall totalling over half of the total precipitation in TV. It is important to note that there is likely some bias due to the noise level of the instruments near 0.5 mm w.e., which is often the total accumulation for one snowfall event.

BOYM (Figure 4d) has consistently low seasonality. Individual, large (> 2 mm w.e.) snow events can govern the season-scale fraction of snowfall for that year like we see in 2007 where Summer precipitation (9.5 mm w.e.) is nearly double that of Spring (5.0 mm w.e.). Low average precipitation and the occurrence of large snow events is likely responsible for its large interannual variability. We do not see any season-specific trend in snowfall at BOYM. HOEM (Figure 4c) also does not show any trends in snowfall for any season although data availability is limited to the last decade.





Toward the coast, FRLM exhibits greatest seasonality and the greatest frequency of anomalously high precipitation (Figure 4b and f). Trends in light season precipitation are governed by Spring and Fall snowfall. Prior to 2009, FRLM received consistently high precipitation during the Spring and Fall, and frequently exceeded the seasonal means. In 2007, precipitation during the Spring was 15.5 mm w.e. higher than the mean. Following 2007, FRLM had reduced seasonality and snowfall in

general. While light season precipitation in FRLM appears to follow *c*. 5-7 year trending patterns, seasonal-scale observations suggest the controls vary on sub-annual timescales.  The stations are not predictors for each other in general, however FRLM and BOYM do exhibit some similarities during seasons with anomalously high snowfall. Precipitation during Spring at FRLM indicates high Summer snowfall at BOYM ($r^2 = 0.89$; $p < 0.001$), especially during high precipitation seasons like 2004, 2007, 2009, and 2012.

**3.2 Snow Persistence**

Data from the camera at Lake Hoare reveals that the average annual snow cover days tripled from $37 \pm 14$ to $106 \pm 15$ days following the 2011 snow year. A time-based snow cover heatmap (Figure 5) suggests that the increase in days with snow cover is primarily due to an increase during the Spring and Fall.  Missing data from the Spring and Fall of snow years 2006, 2010, 2011, 2016, and 2017 may inaccurately portray low snow cover for those seasons. The increase in snow cover

may have been more gradual. Annual snow persistence was derived from dividing snow cover by the number of events. When broken down by season (Figure 6), it is clear that snow persistence is increasing, but only during the Fall. On average, Fall snow cover lasts 3 days longer than Spring and 10 days longer than Summer. Fall of 2017 had an anomalously high persistence due to precipitation on March 1 that lasted through the final sunset. The Mann-Kendall trend test indicates Fall persistence is increasing by 1.1 day each year ($p < 0.01$). Empirical relationships rule out increased volume and more snowfall events as the

drivers of the increase in Fall persistence, which suggests it is likely climatic.

Linear regression analysis shows accumulation at Commonwealth Glacier stake 23 and the percentage of days with snow cover at Lake Hoare are strongly correlated ($r2 = 0.74$; $p < 0.001$) (Figure 7a). The long-term record reveals that summer snow cover at Lake Hoare is highly variable (Figure 7b). Multi-year trends are present prior to 2005, followed by increased interannual variability up until 2013. The most recent 5 years of data show an increase in the fraction of days with snow cover.

An even longer record is required to determine if summer snow cover is increasing on longer timescales or if it will decline again, similar to the 4-5 year trends of increasing and decreasing snow cover prior to 2006. The incorporation of other climatic factors would likely improve the snow cover estimates, however, for the purposes of this study accumulation on Commonwealth Glacier is a sufficient proxy for understanding variability in percent days with snow cover at Lake Hoare.

**3.3 Data Processing and Type Intercomparison**

Comparison with the previously published record allows for interpretation of AWS data in a historical context (Figure 8). Overlapping measurement types agree within about 7 mm w.e. across all stations. Data quality at HOEM do not allow for



overlapping data with the historical record. Snow derived from the sonic ranger only overlaps with previous measurements twice at FRLM and both measurements agree within 3 mm w.e. Agreement is variable at EXEM and the difference ranges from 0.5 mm w.e. (2001) to 31 mm w.e. (2004). The greatest disagreement in data processing between studies is with the weighing bucket gauges, especially in 2003 at EXEM and 2004 at both EXEM and BOYM where the Fountain et al. (2010)

measurements exceeded those in this study by 15 to 30 mm w.e. The difference is attributed to wind data processing. Recently daily average wind speeds were re-calculated (Obryk et al., 2020). Fountain et al. (2010) did not describe the specific calculations used to derive maximum and average daily wind speeds for data processing. At EXEM, one snow event shows up as 8 mm w.e. on August 3, 2003 which Fountain et al. (2010) includes because the mean wind speed is less than 5 m s$^{-1}$. However, reprocessing of the wind data shows mean wind speed of 17 m s$^{-1}$, which is much higher than the wind speed

threshold of 5 m s$^{-1}$. There is another similar instance in 2004 around May 15 and June 8 where 33 mm w.e. total were included by Fountain et al. (2010) and would have been removed with new wind speed considerations. Across all stations and measurement types, data processing at BOYM shows the greatest agreement (± 2 mm w.e.) with Fountain et al. (2010) likely due to the infrequency of precipitation events and low snow cover which would reduce the possibility of erroneous accumulation measurements.

The sonic ranger both under- and overestimates annual accumulation recorded by the weighing bucket gauge. Sonic ranger measurements only exceed weighing bucket gauge measurements by up to 1 mm w.e. which is within the error of the instrument although it may underestimate accumulation by as much as 10 to 31 mm w.e. The underestimate at BOYM from 2008 is attributed to two 4 mm SWE events recorded by the weighing bucket gauge on January 10 and 11. The camera confirms that Lake Hoare received a few centimetres of snow on both days which disappeared the same day. Annual accumulation

calculations from sonic ranger data based on daily averages may underestimate years with events with persistence less than 1 day.

Reanalysis of precipitation in terms of a snow year, yields annual precipitation ranging from 1 to 43 mm w.e. with an annual average precipitation of 8.5 mm w.e. Pettitt's test reveals a changepoint in 2011 at FRLM (p < 0.05). It also identifies a changepoint in 2007 at EXEM, although it is not statistically significant (p < 0.1). Across all stations, from 2004 to 2017,

the Pettitt test identifies a changepoint in 2009 (p < 0.05). From the Mann-Kendall test, the only significant trend is increasing average precipitation in Taylor Valley by 3 mm w.e. a$^{-1}$ (p < 0.05) prior to 2009. Following 2009, the Mann-Kendall test reveals decreasing precipitation by 1 mm w.e. a$^{-1}$ although the trend is not statistically significant (p = 0.06).

## 4 Discussion

### 4.1 Measuring Precipitation in Taylor Valley for Ecosystem Monitoring

Measurements from ultrasonic depth sensors alone are insufficient for recording meaningful snowfall in Taylor Valley during summer months because the dark ground is inherently warmer than icy surfaces, leading to less accumulation and higher



ablation rates compared to icy surfaces of lakes and glaciers. Estimates of the impact of snow on sub-ice ecosystems may be low because the recorded accumulation on land surfaces may not reflect accumulation on the colder icy surfaces. The low signal-to-noise ratio and instrument sensitivity to high wind speeds mean that light, low persistence snowfall recorded by ultrasonic depth sensors may be filtered out with current, necessary data processing methods. If snow both accumulates and

ablates within one day, then the daily average accumulation is likely about half of the actual accumulation recorded by the sonic ranger. Depending on the rate of accumulation and dissipation, precipitation from ultrasonic transducers could be underestimated where accumulation is derived from the difference in daily averages. High wind speeds in TV require conservative data filtering based on climate conditions presumed to be unrelated to snowfall (Fountain et al., 2010), so it is important to couple ultrasonic depth measurements with other measurement techniques.

10       In addition to the limited ability of ultrasonic sensors to pick up low persistence snow events, differences between ultrasonic and weighing bucket gauge measurements could also be explained by the variability in snow density. The weighing bucket gauge measures w.e. directly, whereas measurements from the distance ranger are converted to w.e. using the same density ($83$ kg m$^{-3}$) for all precipitation measurements. Fountain et al. (2010) compares automated precipitation measurement types in detail and note their sensitivity to high wind speeds. High wind speeds increase the noise level of the ultrasonic and

can shake the weighing bucket gauge, both appearing as false precipitation events.

      We reconstructed percent days with snow cover at Lake Hoare from accumulation stake data measured in the accumulation zone of Commonwealth Glacier and identified a shift to increased interannual variability from 5-year trending patterns beginning in 2005. This shift coincides with a shift in mean annual air temperature at Lake Hoare from cooling to no trend (Obryk et al., 2020). While the reconstructed shift in snow cover appears to coincide with the changing temperature

trend, it may in fact reflect a change in glacier processes. Hoffman et al. (2016) developed an ablation model for all of Taylor Valley glaciers from ablation stakes in the accumulation zones. The model regularly underpredicted melt following 2006, but was corrected by reducing the albedo of the toe of the glacier. The model and stakes focused on the ablation zone rather than the accumulation zone and the glacier accumulation record may accurately represent snow cover trends at Lake Hoare although a longer record is necessary to confirm the long-term viability of the relationship. It is important to note that the reduced albedo

is likely due to increased sediment load (Hoffman et al., 2016) or surface roughness (Bergstrom et al., 2020), and not reduced snow cover.

      Although this study focuses on snowfall measurements, snow cover may actually play a greater role than snowfall in influencing the ecology of Taylor Valley. Strong winds convey snow from mountain peaks to the valley floor. Those events are removed because of the reduced accuracy of the automated measurements during that time which could lead to an

underrepresentation of snow accumulation at the valley-floor. The most reliable way to measure snowfall and snow cover for ecosystem monitoring in Taylor Valley, a primary goal of the MCM LTER, is to couple ultrasonic or weighing bucket gauge measurements with daily imagery.



Snow persistence inferred from the camera at Lake Hoare may overestimate persistence at other locations in Taylor Valley. Evelyn et al. (2013) mapped aerial ablation rates throughout TV over the 2009-10 snow year. They found that from October through January, Lake Hoare had the lowest relative loss of snow-covered area (72%) when compared to Lake Fryxell (93%) and Lake Bonney (97%). Lake Hoare lies in a narrow portion of Taylor Valley and receives the least radiation annually

(Dana et al., 1998; Acosta et al., 2020). It is important to also note that Lake Fryxell receives an average of 8 mm w.e. more snowfall than Lake Hoare which may buffer reduced persistence associated with climatic conditions. Additionally, the spatial and temporal pattern of greatest snow volume near the coast during Spring and Fall coupled with greatest persistence and increased snow cover during Spring and Fall at Lake Hoare inherently suggests snow likely plays the largest role nearest the coast and during Spring and Fall, making coastal snow cover monitoring increasingly important.

**4.2 Drivers: Sea Ice and Climate**

Regression analysis indicated no correlation between snow cover in TV and sea ice extent in McMurdo Sound. This is consistent with the findings from Fountain et al. (2010) where they saw a lack of correlation with distance to sea ice edge. Lack of correlation with sea ice extent and temperature contradicts what would be expected since less sea ice and higher temperatures would increase the potential moisture source for precipitation as well as increase RH which would increase snow

persistence. It is important to look instead at seasonal-scale precipitation since tropic-Antarctic teleconnections influence the source of air masses in the Spring and Summer with minimal effect in the Fall and Winter. The discrepancy may be explained by interannual variability of atmospheric patterns that control moisture transport pathways described by Patterson et al. (2005). Fogt and Bromwich (2006) showed tropic-Antarctic teleconnections are strongest in Spring and Summer and weak in Fall and Winter. Low average precipitation means one event could tip the annual accumulation above or below the average, so the

variability associated with ENSO may overwrite the sea ice signal. Because nearly a quarter of the snowfall occurs nearest the coast during the Fall the effect may be better represented from more coastal photographs, which are not yet available.

Snowfall in Taylor Valley is controlled by multiple atmospheric phenomena that govern moisture transport pathways, sea ice extent and regional climate variability. Low precipitation volume makes it difficult to identify trends and drivers in TV because of the potentially strong influence of individual events relative to annual precipitation. Additionally, the longest record

at Lake Bonney has yet to exceed the 30-year mark for defining climate in a region. We can, however, begin to make inferences about the steady increase in snow persistence during the Fall, and to a lesser extent Summer, at Lake Hoare. Atmospheric phenomena have a seasonality and tropic-Antarctic teleconnections are insignificant in the Fall, which suggests the increasing persistence may be indicative of the changing climate.

Positioning in the central portion of the valley means Lake Hoare experiences both continental and coastal climate

influences (Lyons et al., 2000) and persistence trends may not translate to other portions of the valley. Lake Hoare experiences a seasonal transition from continental to coastal climate as the summer temperature gradient is established (Obryk et al., 2020; Lyons et al., 2000). The relationship of high Spring snowfall at FRLM indicating high Summer snowfall at BOYM may be



due to the annual expansion of coastal climate further inland during the summer. To fully understand how persistence trends at Lake Hoare relate to other areas of the valley, it is important to monitor snow persistence elsewhere in Taylor Valley.

## 4.3 Implications for Hydrology and Ecology

Snow persistence could be increasing because of reduced melt, sublimation, evaporation or a combination of the three. It is
important to determine the contribution of each process because of the different effect they have on the hydrology. Harpold and Brooks (2018) show that under warming conditions, low-humidity regions will experience greater evaporation and sublimation when compared to high-humidity areas which will experience greater melt. Sublimation is the greatest contributor to ablation of snow (Gooseff et al., 2011). Under these assumptions, reduced snow volume and increased snow persistence will further reduce the soil moisture contribution of snow which could have mixed effects on subsurface ice and soil
communities. While there would be less melt to recharge subsurface ice, the increased duration of snow cover could act as a buffer and slow ablation. Soil ecosystems could experience reduced soil moisture from a lesser snowmelt contribution and reduced ablation of subsurface ice.  Surficial communities, however, could experience greater soil moisture from increased ablation of subsurface ice (Wall, 2007).

      More snow cover increases the albedo of Taylor Valley and reduces glacier melt and therefore streamflow. Chinn
(1979) showed heavy early season snowfall reduces streamflow of the Onyx River into Lake Vanda in nearby Wright Valley. Absence of streamflow into Lake Vanda is not directly indicative of no glacier melt, but that evaporation along the river exceeded discharge. Streamflow aides in the seasonal melting of lake ice margins or "moats" and increased snow cover could slow moat development and therefore affect benthic primary producers residing below the seasonal ice-cover. Snow cover is a major contributor to attenuation of light in ice-covered lakes (Hawes 1985; Bolsenga et al., 1996).  Accumulation of snow
on the lake ice could also encourage thicker ice covers by reducing ablation rates and further reduce underwater photosynthetically active radiation. Doran et al. (2002b) note reduced underwater irradiance and a reduction of primary productivity by 9% in the east lobe of Lake Bonney from increased the ice cover thickness. Persisting snow covers could have the potential to create lasting changes to aquatic ecosystem dynamics of TV.

## 5 Conclusions

Our record shows a clear increase in snowfall between 1995 and 2009 of 3 mm w.e. a$^{-1}$, however, snowfall has been decreasing by 1 mm w.e. a$^{-1}$ through 2017, which contradicts the expected to increase of snowfall in polar coastal regions under warming conditions (Bintanja and Selten, 2014).   From 2010 to 2017, annual snowfall in Taylor Valley ranges from 1 to 30 mm w.e., The greatest volume of snow falls nearest the coast during the Spring and Fall and decreases further inland and during the Summer.            Statistical analysis revealed a new spatiotemporal relationship between coastal (FRLM) and continental
(BOYM) precipitation patterns and indicates that high spring snowfall at FRLM likely indicates high summer snowfall at

BOYM. Seasonal-scale variability in snowfall underscores the role for atmospheric teleconnections in controlling precipitation in Taylor Valley. Synoptic-scale atmospheric conditions overshadow the effect of reduced sea ice conditions increasing precipitation.

Accumulation typically does not exceed 0.5 mm w.e. in one snowfall event. Snow cover may surpass snow volume in its ecological impact. Snow cover regulates primary productivity and hydrologic connectivity within Taylor Valley. The fraction of days with snow cover during the light season tripled at the Lake Hoare after 2011, mostly during the Spring and Fall season. Additionally, snow persistence has been increasing by *c.* 1 day each year in the Fall since the start of the record. A continued increase in snow cover and persistence would have both ecologic and hydrologic implications related to the radiation budget in Taylor Valley that contradict the increase in hydrologic and therefore ecologic connectivity predicted by the MCM LTER (Wlostowski et al., 2016).

Continued monitoring of both snowfall and snow cover in Taylor Valley is necessary to fully understand the local response to global climate change, especially since precipitation is controlled by moisture transport pathways whereas persistence reflects climatic conditions *following* a snowfall event. Ultrasonic and weighing bucket gauge measurements should be coupled with photographs to most effectively measure snowfall and snow cover for ecosystem studies in polar desert environments.

## Author Contributions

M. Myers processed all of the data and conducted statistical analyses. Doran and K. Myers were responsible for data processing methods and the inclusion of statistical analyses. Doran and K. Myers also aided in field data collection.

## Competing Interests

The authors declare that they have no conflict of interest.

## Acknowledgements

The authors wish to thank the contractors and Petroleum Helicopters, Inc. for their support in the field. Thanks are also extended to field parties for reporting snowfall observations during the field season. This research is funded by the National Science Foundation Grant #OPP-1637708 for Long Term Ecological Research. Support was also provided by the Louisiana State University John Franks Chair Fund. Any opinions, finding, conclusions, or recommendations expressed in the material are those of the authors and do not necessarily reflect the views of the National Science Foundation.





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





Table 1. Mean seasonal precipitation recorded at each weather station. Error is based on the standard error of the mean. Seasons with less than 75% data available are excluded.

| Weather Station | Mean Precipitation (mm w.e.) | | |
|---|---|---|---|
| | Spring | Summer | Fall |
| Explorer's Cove | 4.4 ± 1.1 | 1.0 ± 0.4 | 3.8 ± 1.1 |
| Lake Fryxell | 5.4 ± 1.6 | 0.5 ± 0.2 | 4.4 ± 1.9 |
| Lake Hoare | 1.7 ± 0.1 | 1.1 ± 0.1 | 1.2 ± 0.2 |
| Lake Bonney | 1.8 ± 0.1 | 1.0 ± 0.1 | 2.1 ± 0.3 |



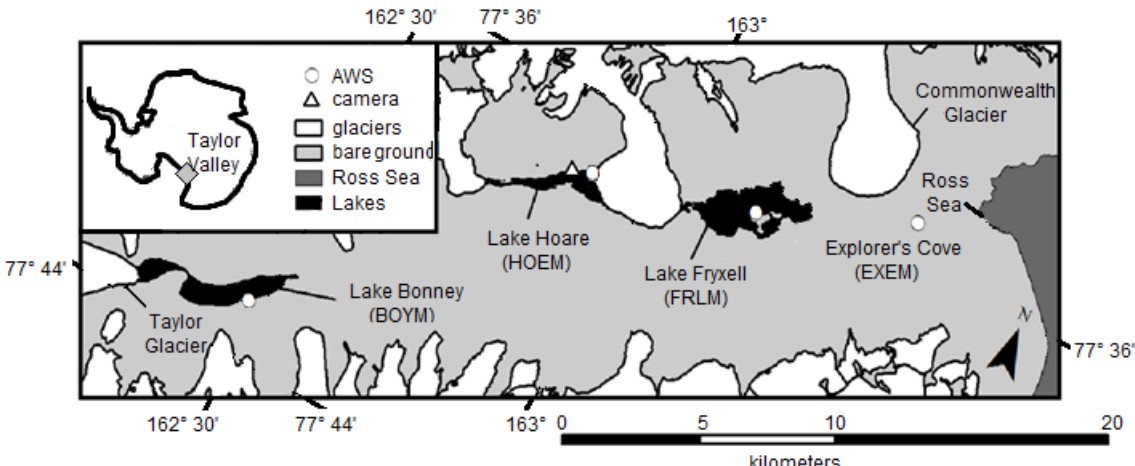

Figure 1. Location of Taylor Valley in Antarctica (inset) with AWS and camera sites as well as station abbreviations indicated.

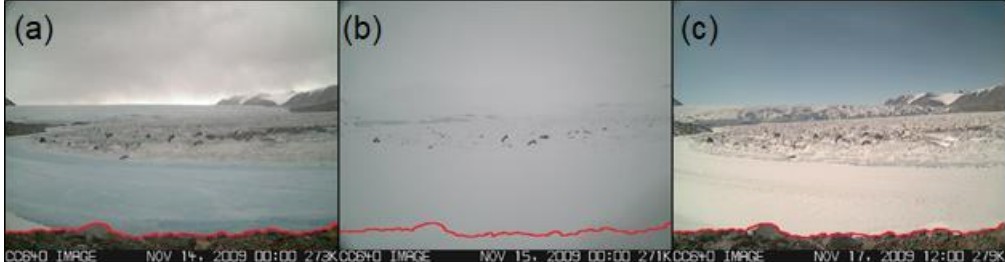

Figure 2. Frames from the camera on the north shore of Lake Hoare facing east toward Canada Glacier and the coast. The ground used to derive snow cover is outlined in red. Lake Hoare is covered by the blue perennially melted moat and permanent rocky ice cover. The snow event captured here has a persistence of 3 days, which is the amount of time between the last image before snow was seen on the ground (a) and the first image where there was no longer any snow on the ground (c).





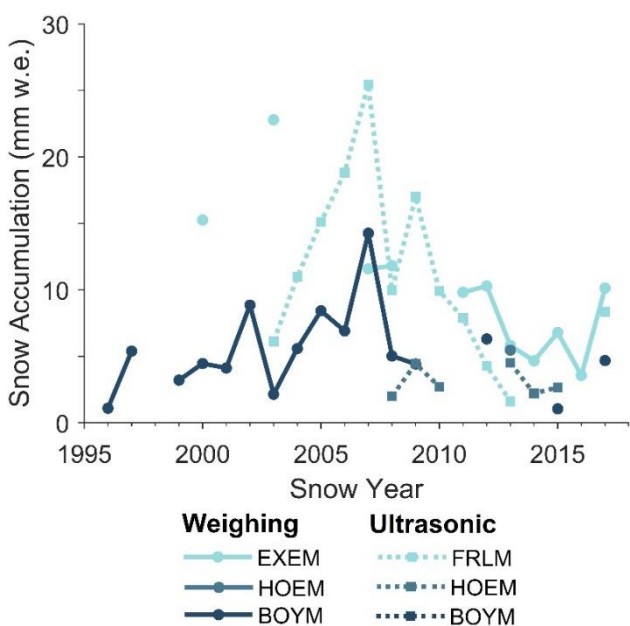

Figure 3. August through April precipitation at each station recorded by ultrasonic distance ranger and weighing precipitation gauge. Snow years with over 75% of data missing are excluded. Station locations are provided in Figure 1.

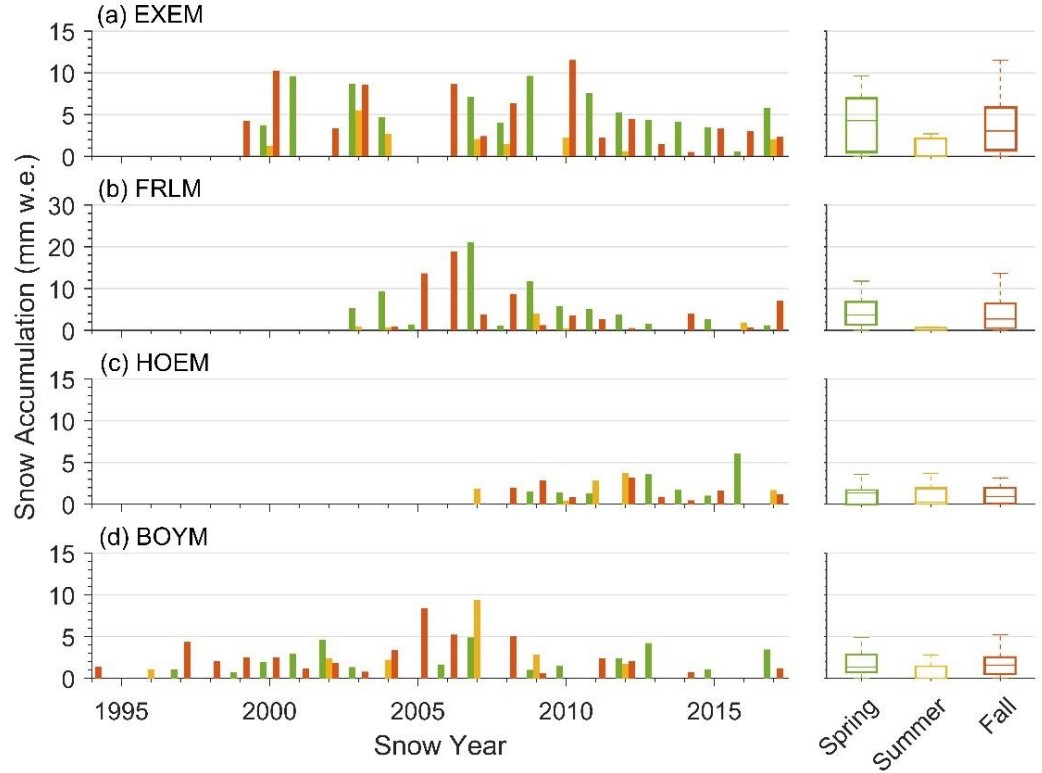

Figure 4. Snow accumulation (mm w.e.) recorded at all stations followed by box plots showing the data distribution. Data are separated by season which is indicated by color. Plotted values can be found in supplementary Table S1 which also differentiates between missing data and 0 mm w.e. accumulation. Weighing bucket data are shown for (a) EXEM and (b) FRLM. Note the difference in y-axis scale in (b) FRLM. Ultrasonic data are shown in (c) HOEM and (d) BOYM.


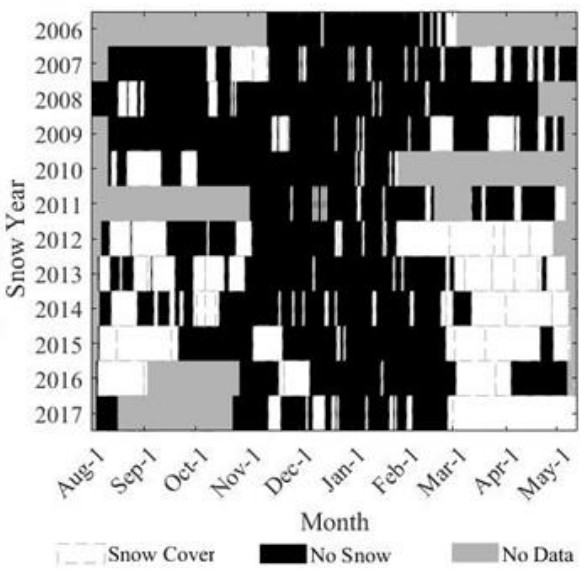

Figure 5. Snow cover record derived from the Lake Hoare camera. Dashed lines around white boxes denote individual events.





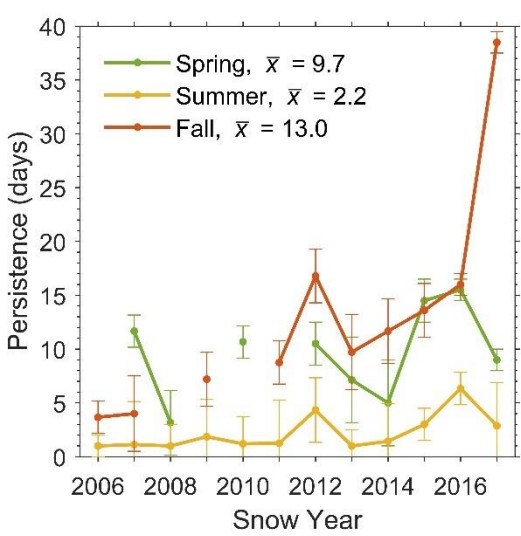

Figure 6. Snow persistence derived from the Lake Hoare camera. Seasons with less than 75% data available are excluded. Error bars are representative of the resolution of snow cover photographs (± 0.5 day per event). Season means are indicated in the legend.





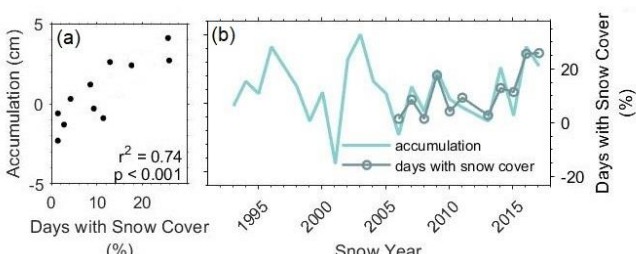

Figure 7. (a) Percentage of days with snow on the ground between November and January plotted against accumulation measured between the same dates on Commonwealth glacier. (b) Timeseries of accumulation and snow cover record.





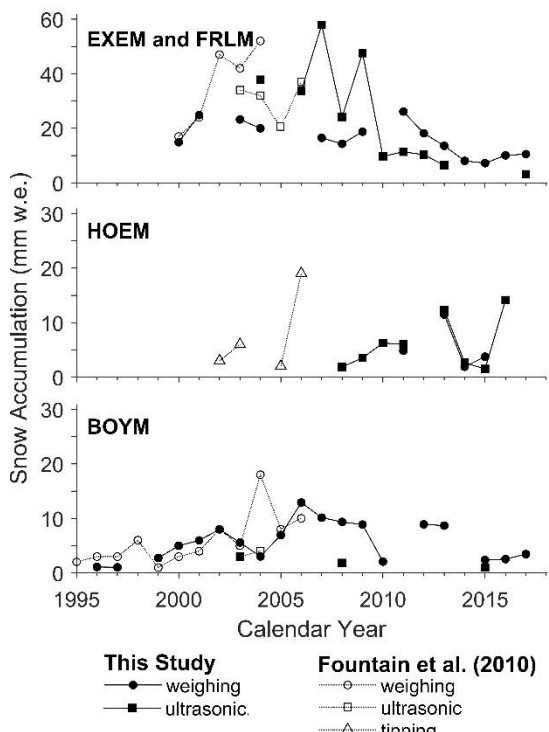

Figure 8. Snow accumulation by calendar year from Fountain et al., 2010 and this study. FRLM only has a sonic ranger and EXEM a weighing bucket gauge. The two are combined in the top plot. Years with less than 75% data available are excluded. Supplementary table S2 contains a table of errors for accumulation derived in this study.

