# Peer review of "Summer valley-floor snowfall in Taylor Valley, Antarctica from 1995 - 2017"

_The Cryosphere, 2020_

## Referee Comment (RC1) · Anonymous Referee #1 · 1 Oct 2020

This manuscript describes updated datasets on snow depth and snow cover in the Taylor Valley. The authors suggest these data may be used for trend analysis and to test hypotheses of weather patterns and forcing thereof. The manuscript is reasonably written and easy to follow, though further care and proof-reading to correct typographical errors would be useful. The dataset is certainly useful and analysis important for describing the snow patterns within the Taylor Valley. However, the manuscript lacks depth for publication as a research article. There is no clear objective other than to use data to re-analyze trends previously observed in the data. While I certainly see the value of these data and the analysis, I think the manuscript would be more appropriate as a data paper in another journal (because The Cryosphere does not offer a data paper format). Below are minor comments.

[Figure]

Page 2 Line 9: is this temperature supposed to be negative?

P3 L15-20: The specific objectives are not very clear. It could help a reader determine precisely what the paper is trying to accomplish if these are re-written for better clarity

P4 L11-12: this is the first time this is phrased as a hypothesis, please state this in the introduction, this would help clarify objectives previously mentioned.

P6 L22: r2

P7 L15: what do you think the potential error in depth conversion to w.e. using a single density observation is? I understand the need for this method, but what magnitude of uncertainty does this bring with it?

P8 L12-13: So how much does density vary in the available observations?

P8 L28: How much do you think make it to the valley floor from the mountain peaks? I would think that so much sublimation would occur in the dry air and high wind speeds that little to no snow would make it that far. Just a thought.

Conclusions: A lot of these conclusions read more like discussion points. I suggest simply stating the conclusions directly.

Figure 3: Could you use colors that contrast a little more, please? It is very difficult to see the differences between the line colors.

Figure 4: It took me a bit to figure out the color symbology. A legend for the colors would be a great help.

Figure 7: What is the r2 and p-value for? Can you show the line for the regression?

---

## Referee Comment (RC2) · Anonymous Referee #2 · 30 Oct 2020

This paper addresses an important subject of the temporal and spatial patterns of snowfall in a polar desert in Antarctica, with implications for assessing climate change and relevance to local ecosystem processes. The paper essentially updates and expands the record of snowfall in Taylor Valley. It is a worthy effort and they add a new measure - persistence of snow cover.

While I heartedly endorse the publication of the report, it has many important flaws that need to be corrected. Overall, the science is fine, but the writing needs major major improvement, as indicated by my extensive comments below.

In short, the entire manuscript has to be rewritten paying close attention to grammar, flow, and definitions. A number of broader issues stand out.

[Figure]

1. Precipitation and snowfall are used interchangeably. No rain falls in this region so why not just use snowfall? By including both words, sometimes in the same sentence, the reader gets the impression that rain is ignored.

2. Some care should be taken to distinguish between snowfall and snow accumulation because they are different. Also snow cover is a bit ambiguous and is typically used to refer to snow at a point rather than across a landscape, except at the end of the report then its used to mean the latter. I think the use of snow cover can be avoided except for meaning across a landscape.

3. Often the authors refer to snowfall volume, yet they use one dimensional units of cm. Normal practice is to refer to precipitation amounts as depth in mm or cm. I suppose the authors could refer to specific volume, which also has units of mm or cm, but why complicate things. Unless they want to calculate volume of snow in a watershed, I'd stick with depth.

4. One important issue that is glossed over is uncertainty. If the data are compared then uncertainty needs to be included or the comparison has no context. Assessment of uncertainty, of both snowfall depth and duration, should be addressed in a separate paragraph.

5. Finally, I have a problem with the notion that one station can predict the snowfall at another station the following season. Do the authors think there is a teleconnection extending the 10-20 km between stations? By what physical process explains this phenomenon. Why aren't other station pairs predictors? How well are the stations correlated? How are we to know that this predicator is not a spurious correlation?

Detailed comments

Line Comment

This is only part of the reason. The other part is the strong rain shadow exerted by the TransAntarctic Mountains. See, Monaghan AJ, Bromwich DH, Powers JG and

Manning KW (2005) The Climate of the McMurdo, Antarctica, Region as Represented by One Year of Forecasts from the Antarctic Mesoscale Prediction System*. Journal of Climate 18(8), 1174–1189 and see, Fountain et al., 2010

This reference is for sea ice, a more local refence, highlighting the effects on runoff from glaciers in the Dry Valleys, including the energy balance causes, is needed

A more apt citation here, based on the physics of energy balance is Hoffman et al., 2016, already included in the reference list.

"They excluded. . .." It is not clear whether this study also excluded windy events or not.

Abrupt and confusing transition from automatic measurements to a brief study of snow? It is unclear what you are trying to measure, not density, that is assumed to be 83 kg/m3, but then density was measured. Please clarify

"with Winter excluded for the same reason" Same as what? "Spring begins with first light. . ." When is that?

"ends with final sunset" When is that? "Dates coincide with statistically distinct climate conditions" Climate is a very broad umbrella, which conditions, specifically?

'Instrumental' rather than "meteorological"?

Why Commonwealth Glacier rather than Canada Glacier, which is adjacent to Lake Hoare? And why stake 23>

"Precipitation". Is this a better descriptor or is snowfall? There is no rain and all precipitation is snow, so should the general term be used? I think not, it has a vagueness

to it that is unnecessary.

"Again"? When was it focused previously? Also considering that the seasons are partly defined by first and last light, if I understand the methods correctly, why is a 'light season' necessary? It's confusing. Also, this sentence underscores the confusing issue between precipitation and snowfall. Here you, 'focus on snow...', but the subheading is about precipitation, so the inference is rain is being ignored.

2007? Does this coincide with the results of Obryk on temporal break points?

This sentence doesn't seem logically connected. How is seasonal variability connected with differences in atmospheric influences? What influences are being considered?

"spatial control" or spatial difference? Controls at this point in Results are unknown, but differences are known.

You don't really mean 'volumes' right? Then it begs the question over what area are you measuring the volume. For precipitation, depth is the normal dimension used. Furthermore, the units of mm w.e. are not volume but a linear distance, so the dimensions of volume are wrong. Also, are these values averages? Please clarify. If yes, what is the standard deviation?

If a third of w.e. snow occurs in spring and another third occurs in autumn (two-thirds, not 'totaling over half', then one-third occurs in summer. If this is right, it doesn't square with the measurements at either the coastal or inland stations. Or am I confused?

"Bias". This brings up a couple of good points. First, do the authors mean 'bias' as in the measurements tend to be too high or too low? Second, do the authors mean 'uncertainty'. Considering they are comparing values, to make the comparison meaningful, they need to report uncertainty.

Looks like HOEM has a consistently lower seasonality than BOYM. Or is my interpretation due to missing data?

Revise, "where Summer precipitation (9.5 mm w.e.)   is nearly" to "when Summer….was nearly".

Again, another example of precipitation vs snowfall, "Low average precipitation and the occurrence of large snow events". So little rain and large snowfall?

Looks to me that EXEM has the greatest variability, not FRLM. Is that due to missing data at FRLM? It looks like prior to 2003 FRLM received no snowfall.

"stations are not predictors". I assume the authors do not mean the stations are correlated because in the following sentence Spring snowfall at FRLM predicts high summer snowfall at BOYM. I would have thought that a correlation matrix between stations would be included or perhaps referred to in supplementary data to support the notion that the station snowfall is not correlated between stations (is this right?). There is no physical reason for one station to predict snowfall at a later date at another station, unless it does so at the same station. Given the stations are only few km apart the prediction is not based on the movement of air mass systems or a teleconnection. Its just persistence in the system.

"snow cover heatmap"? Awkward, revise.

"may inaccurately portray low snow cover for those seasons". It can't portray low snow cover, because like you say, the data is missing.

"more gradual" than what? No rapid increase was identified.

replace 'high' with 'long'

"snow cover at Lake Hoare is highly variable', clarify, snow cover persistence?

Delete the last two sentences in the paragraph, they don't say anything.

Replace 'ground' with 'soil' or 'rock and soil'

Delete the sentence about sub-ice ecosystems, it doesn't go anywhere and is a distraction from the subject of the paragraph snowfall versus accumulation. If the ecosystem issue is important, develop in a separate paragraph.

Delete 'necessary'. I'm sure one could figure a work-around if needed.

Replace 'dissipation' with 'ablation', replace 'from' with 'based on'

Replace 'pick up' with 'detect'

The last two sentences sort of repeat the last sentence in the previous paragraph. Can the influence of high winds be more fully addressed in a single paragraph rather than partly in two paragraphs?

" from cooling to no trend" Awkward. Rather than a 'shift', how about "a changing trend from cooling to no trend"?

Delete the rest of the paragraph starting with, 'While the reconstructed. . .' The ablation and melt in the ablation zone vis a vis Hoffman et al., 2016 has nothing to do with snow cover. The text awkwardly summarizes the model and at the end of the paragraph, the authors back into a suggested process, reduced snow cover. Hoffman et al., do not argue that the lack of snow causes increased melt, they argue increased sediment on the ice surface. So, I don't know the purpose of these sentences.

Delete this paragraph, it doesn't make much sense. It starts to make an argument for snow cover vs snowfall relevance to local ecology. But that subject is dropped, and the subject shifts to the importance of winds again. It ends with an unsubstantiated statement about the best approach to measure snowfall and snow cover. Strangely, the ecosystem argument ignores an important aspect of snowfall and snow cover, its spatial distribution. In any case this paragraph doesn't really say anything important to the paper.

"lowest relative loss of snow-covered area (72%)" What does this mean? And what is 72%? Does this mean that the area of Lake Hoare lost 72% of its snow cover? On average? Or is 72% of the Lake Hoare area covered in snow?

"least radiation" nuclear radiation?

"may buffer reduced persistence associated with climatic conditions" Vague. What associated climate conditions?

volume, not depth? And the last sentence is very confused. How does snow at Lake Hoare inform on snow at the coast? "snow likely plays the larger role. . ." in what? And makes its monitoring increasingly important? Why not important, why is it increasing in importance? I might argue that it is more important to monitor snow up valley where there is less moisture.

Delete section 4.2? It doesn't come to any substantial conclusion. Given that the region is a desert and one large snow event can change the season of maximum snow fall, clearly the statistics will be very noisy and regressions and teleconnections will be insignificant. If the authors feel that this section is important, reduce it to one small tight paragraph.

Delete 'sea ice extent' In the previous paragraph it was shown to be irrelevant to snowfall.

Delete section 4.3. This section lightly argues for the relevance of snow to the hydrology and ecology of Taylor Valley. Unfortunately, its not particularly insightful and the topics have been better covered by the authors in the introduction. Also, there are

several conceptual mistakes in 4.3 summarized below.

"high-humidity areas which will experience greater melt" These areas will melt? Surely you don't mean that. Do you mean snow in those areas?

"Sublimation is the greatest contributor to ablation of snow" Not true, in most temperate regions of the world melt is the biggest factor with sublimation playing a very small role.

"Under these assumptions, reduced snow volume and increased snow persistence will further reduce the soil moisture contribution of snow which could have mixed effects on subsurface ice and soil communities. While there would be less melt to recharge subsurface ice, the increased duration of snow cover could act as a buffer and slow ablation." These two sentences are wrong showing a misunderstanding of the heat and mass transfer of snow over soil, particularly a relatively warm snow over much colder permafrost at depth.

The predictive capability of high spring snowfall at FRLM to indicate high summer snowfall at BOYM only a short distance away is odd. This appears to be a case of correlation without causation, and not examined carefully by the authors. Furthermore it is very odd that no other station pairs show this, which makes me think this is specious and not worthy of inclusion in the conclusions.

This sentence is unsubstantiated by anything prior in the report and should be deleted.

This paragraph should be deleted it doesn't say anything substantial.

Figure 1. I've always thought that no acronyms should be included in a figure without explanation, otherwise, the reader has to search the text for interpretation. I recommend AWS be spelled out too.
Figure 2. This is a confusing figure. The caption says the monitored area is outlined in red, but I only see a red line, not a polygon, so no 'area' is outlined. After some inspection I realized it was at the bottom of the photo. It would help the reader if the photo was cropped to minimize much of the sky in order to emphasize the monitored area. 4 'perennially melted moat'? It is always melted? The moat is part of the lake? Unclear The last sentence is awkward, please revise for clarity

Figure 4 I think it is important here to show missing data. Otherwise the plot is misleading, no bar is interpreted as zero snowfall. For example, it appears HOEM had snow snow accumulation between 1994 and 2006. The bar graph to the right, are these averages? If so, what is the sample size of each? It bears on the statistical differences between seasons and between stations.

Figure 6. In the legend the mean is indicated by X-bar. But X-bar would be a snow year. Persistence is on the y axis, Y-bar? Why is the resolution of the photo +/- 0.5 days? This was not explained in the Methods.

Figure 7. (b) where is the scale for accumulation? The tick marks suggest a scale different from (a).

---

## Referee Comment (RC3) · Anonymous Referee #3 · 16 Nov 2020

General comments and recommendation

This paper aims at describing snowfall and snow cover data that has been sampled in the Taylor Valley, in Antarctica. The protocol and the stations used to get the measurements are correctly described, and acquiring such data in this remote area is clearly a huge effort. The paper is well presented. However, the investigations based on these data are superficial, because many conclusions are not based on solid investigations, and some of them appear speculative. I would recommend to either publish this dataset in a web interface/journal with a DOI reference, or to conduct more investigations to prepare a manuscript. In such a form, I would recommend to reject this article.

There are two major points for which the study is not appropriate to make a scientific

article to my point of view:

* The trend analysis is based on very short timeseries, a point that strongly limits the possibility to evidence any climatic trend in the area. To my point of view, however, it would be interesting to provide a study focusing on the interannual variability. Such a study would require considering more variable/processes than the snowfall rates and snow persistence that are observed by the authors. In particular, the conclusions suggesting climatic signals in this area could be based on temperature/wind/pressure data, using observations at the local scale and potentially reanalysis at the regional scale. This would lead credence to the section devoted to the teleconnections between the polar and the tropical to middle latitude areas.

* The other weakness of this study is related to the links between snowfall and sea ice that are mentioned by the author trough all the article, whereas there is not any sea-ice data in the study. In addition, the authors claim that a sea-ice reduction is expected in this area, that would favour a precipitation increase in relation to more moisture in the atmosphere. Even if such precipitation increase is expected in Antarctica under climate change, the sea-ice did not show any clear trend over the last decades, and even a slight increase in the Ross sea (De Santis et al., 2017). This point should be clarified when preparing a new article.

List of comments

A point-by-point list of comments is provided below, with suggestions that could be taken into account to conduct a more complete study.

Introduction:

* What do you think about extending the area shown in Figure 1? This would allow to evidence that the Taylor Valley is a valley surrounded by mountains/glaciers.

* P.2 L.9: annual mean of air temperature observed on average in TV by Obryk et al. is -20°C, so 18.5°C seems to warm (maybe a – sign is missing?).
* P.2 L.27: the sea-ice extent in the Southern Hemisphere has been increasing over the last decades in particular in the area of the Ross Sea (de Santis et al., 2017), so should we expect a decrease in snowfall? This should be considered in the introduction and all over the manuscript.

Methods:

* P.3 L.25 to L.30: It is claimed that the observation of precipitation is considered only when the wind is not exceeding 5 m.s-1. But is it realistic to consider that there is no local snowfall with stronger winds? When the snow is drifted away with the wind, this does not mean that there is no snowfall, isn't it? I would expect more explanations for the situations when snowfall occurs during windstorms.

* Table 1: Is the uncertainty shown in Table 1 includes the uncertainties of snowfall related to wind impact of sensors?

* P.4, L.2: is the snow density systematically equal to 83 kg.m-3? That sounds like a strong assumption.

* P.4, L.9: "Winter excluded for the same reason" -> which reason? The sentence is not clear. You mean that you do not focus on the winter season because of the lack of sunlight, isn't it? What are the limitations related to this protocol?

* Figure 2: a) and c) are mentioned in the caption, but not b).

Results

* The discussion focusing on the volume of precipitation variability appears speculative, in particular because of the shortness of the time series as well as because of the missing data. The potential links between the spring snowfall at FRLM and the summer snowfall at BOYM is far from being clear visually. Even if the correlation is significant, would it be possible that this happened by chance? I would suggest providing also a power analysis (e.g. Von Storch and Zwiers, 2001) to estimate whether such a significant correlation has been obtained "by chance". The trends computed over such short

periods should be considered very carefully also.

* P.5: Even shown in Figure 1, the names of the stations presented in the results and in particular in Figure 3 should be fully explained/detailed (BOYM, EXEM, HOEM, etc...)

* P.5, L20: what does mean the "c." before 0.5 mm in this sentence?

* P.6, L.2: a reference to Figure 4f is given whereas there is no visible f) in Figure 4.

* Figure 5: the temporal resolution of the heatmap should be specified in the caption (daily resolution?).

* Figure 7: it seems that the number of days are centred over an average value, because there are negative values. This should be detailed in the caption.

* P.7, L22: "precipitation in terms of a snow year" -> Does it mean that the winter period is also included in the annual value? Or is the winter period excluded for the two sets of observation?

Discussion

* P.8, L13: Again, it could be interesting to give an estimation of both the spatial and the temporal variability of the snow density, because the choice of a constant value of 83 kg.m-2 seems arbitrary. Also, it would be interesting to estimate the uncertainty of snowfall rates that directly emanate from the density uncertainty.

* Did you consider to measure drifting snow, like Amory et al. (2020)?

* P.9, L.11: it is claimed that there is no correlation between snow cover in TV and sea-ice extent, but there is neither any figure, neither any number to evidence this finding. This finding should be illustrated with numbers or should appear in a previous publication. Same remark can be done with the temperature observations.

* P.9, L.28: "the increasing persistence may be indicative of the changing climate" -> This sounds very speculative, because the timeseries are very short, with missing data,

the snow accumulation changes have a strong spatial variability as evidenced by the signals that differ from one site to another. Also there is no analysis in the article concerning the mid-latitude to polar teleconnections, and in a general way the article does not include any investigations related to snowfall/snow persistence and atmospheric variables, which led few credence to the hypothesis appearing in the discussion.

Implication for Hydrology and Ecology

* The discussion related to the hydrological consequences is not clear. A situation with reduced snow volume and increased snow persistence (L.8) is pointed out as a situation that would favour a decrease of soil moisture. That's true, but if I have understood this article, I do not see in the previous section such opposite trends for snow persistence and snow accumulation.

* P.10, L8: "Sublimation is the greatest contributor to ablation of snow" -> Could you mention where such finding is applicable, please?

Conclusions

* P.10, L25: "Our record shows a clear increase in snowfall..." -> could you remind where and when did you observe this trend, please?

* P10. "Snowfall has been decreasing trough 2017 [from 2009...], which contradicts the expected increase in snowfall in polar regions under warming conditions" -> To my mind, this sentence is a too-simplified view of the climate change in polar area, because this observed decrease of snowfall occurred only over 8 years (2009-2017), a short period for which the internal variability of the climate system can lead to any change of precipitation because of its chaotic nature, even if it is superimposed with the long-term warming trend observed over the last decades. Similarly, the authors could write an opposite sentence when they describe the increasing trend of snowfall that they observed between 1995 and 2009.

* P.10, L.28: the sentence has a grammatical issue, with a capital letter after a comma,

and maybe a missing verb somewhere, and a blank space located at the middle of the sentence?

* P.11, L2: Again, it is claimed that there is no link between the increasing precipitation and the reduced sea ice, but there is no number evidencing any sea ice retreat, and such number cannot be found in the citation Fountain et al. (2010). Also, an impact of the synoptic-scale atmospheric conditions is suggested, but without any corresponding result shown in the article.

* Data availability: If the article is published, the data used in this study should be made available on a web interface, with a doi reference.

References

Amory, C.: Drifting-snow statistics from multiple-year autonomous measurements in Adélie Land, East Antarctica, The Cryosphere, 14, 1713–1725, https://doi.org/10.5194/tc-14-1713-2020, 2020.

Angela De Santis, Eder Maier, Rodrigo Gomez & Inti Gonzalez (2017): Antarctica, 1979–2016 sea ice extent: total versus regional trends, anomalies, and correlation with climatological variables, International Journal of Remote Sensing, DOI:10.1080/01431161.2017.1363440

Von Storch, H. and Zwiers, F.W., 2001. Statistical analysis in climate research. Cambridge university press.

---

## Author Comment (AC1) · 5 Jan 2021

AUTHOR RESPONSES TO REVIEWER 1 COMMENTARY ON MYERS ET AL MANUSCRIPT

This manuscript describes updated datasets on snow depth and snow cover in the Taylor Valley. The authors suggest these data may be used for trend analysis and to test hypotheses of weather patterns and forcing thereof. The manuscript is reasonably written and easy to follow, though further care and proof-reading to correct typographical errors would be useful. The dataset is certainly useful and analysis important for describing the snow patterns within the Taylor Valley. However, the manuscript lacks depth for publication as a research article. There is no clear objective other than to

use data to re-analyze trends previously observed in the data. While I certainly see the value of these data and the analysis, I think the manuscript would be more appropriate as a data paper in another journal (because The Cryosphere does not offer a data paper format). Below are minor comments.

[M, D, & M] We feel this paper is well suited for The Cryosphere with more attention given to the trend analysis and the uncertainty dependent upon the density used to convert depth to mm w.e. A more appropriate analysis for understanding the relationship of snow to sea ice extent and the position of the Amundsen Sea Low would be beneficial. The manuscript will be greatly improved by the comments from Reviewer 1.

Page 2 Line 9: Is this temperature supposed to be negative?

[M, D, & M] Yes, the temperature should be negative and will be reflected in the revised manuscript.

P3 L15-20: The specific objectives are not very clear. It could help a reader determine precisely what the paper is trying to accomplish if these are re-written for better clarity.

[M, D, & M] We agree and propose to replace the end of this paragraph beginning on line 17: "...temperature shift. First, we extend the record of snowfall from 1995 through 2017 from four valley-bottom AWS. We conduct a changepoint and trend analysis on the snowfall dataset to understand seasonal and annual variability in snowfall in the context of the climate shift described by Obryk et al. (2020). The snowfall record is augmented with a decade (2006 to 2017) of snow cover and persistence data derived from daily photographs taken in Taylor Valley. We focus on seasonal-scale trends to highlight climate variability which may depend on shorter-lived factors such as atmospheric oscillations. Finally, we reveal shortfalls of presently established methods for monitoring precipitation in polar deserts and highlight how daily photographs improve them."

P4 L11-12: This is the first time this is phrased as a hypothesis, please state this in the

introduction, this would help clarify objectives previously mentioned.

[M, D, & M] We agree with this comment. A sentence will be added to the end of the introduction to introduce the hypothesis of sea ice influencing snow in Taylor Valley.

P6 L22: r2

[M, D, & M] The manuscript will be updated.

P7 L15: What do you think the potential error in depth conversion to w.e. using a single density observation is? I understand the need for this method, but what magnitude of uncertainty does this bring with it?

[M, D, & M] Combined answers in next section

P8 L12-13: So how much does density vary in the available observations?

[M, D, & M] Snow density is infrequently observed and reported in the Dry Valleys, but reported average densities range from 80 to 100 kg m3 for the McMurdo area (Keys, 1980) and our field measurement was 83 kg m3. The use of 83 kg m3 in our calculations could result in an underestimation of snow volume by upwards of 20%. This could have implications for our changepoint and trend analysis. A sensitivity test of those results to snow density could benefit the paper in that respect. However, the sonic measurements generally agree with the weighing bucket measurements. Where annual measurements do overlap, the ultrasonic does tend to underestimate accumulation by ∼1 mm w.e. (∼24%). Where the sonic ranger underestimated annual snowfall (Figure 8, BOYM), it was because snow persistence was so short that the event wasn't captured. These comments will be added to the manuscript's discussion.

P8 L28: How much do you think makes it to the valley floor from the mountain peaks? I would think that so much sublimation would occur in the dry air and high wind speeds that little to no snow would make it that far. Just a thought.

[M, D, & M] I would assume not much snow for the same reasons. Any perceived

"snow" by the instruments filtered out during strong winds is probably an artefact of instrument performance and supports our decision to filter them out. Perhaps that statement should read "snow could be conveyed..." and that will be updated. Conclusions: A lot of these conclusions read more like discussion points. I suggest simply stating the conclusions directly.

[M, D, & M] We agree, the conclusion should be updated to reflect the conclusions rather than discussion topics.

Figure 3: Could you use colors that contrast a little more, please? It is very difficult to see the differences between the line colors.

[M, D, & M] We agree and the figure will be updated.

Figure 4: It took me a bit to figure out the color symbology. A legend for the colors would be a great help.

[M, D, & M] We agree. The figure will be updated.

Figure 7: What is the r2 and p-value for? Can you show the line for the regression?

[M, D, & M] We wanted to show the goodness of fit. The regression line will be added to the plot.

---

## Author Comment (AC2) · 5 Jan 2021

AUTHOR RESPONSES TO REVIEWER 2 COMMENTARY ON MYERS ET AL MANUSCRIPT

This paper addresses an important subject of the temporal and spatial patterns of snowfall in a polar desert in Antarctica, with implications for assessing climate change and relevance to local ecosystem processes. The paper essentially updates and expands the record of snowfall in Taylor Valley. It is a worthy effort and they add a new measure - persistence of snow cover.

While I heartedly endorse the publication of the report, it has many important flaws that need to be corrected. Overall, the science is fine, but the writing needs major major

improvement, as indicated by my extensive comments below.

In short, the entire manuscript has to be rewritten paying close attention to grammar, flow, and definitions. A number of broader issues stand out.

1. Precipitation and snowfall are used interchangeably. No rain falls in this region so why not just use snowfall? By including both words, sometimes in the same sentence, the reader gets the impression that rain is ignored.

2. Some care should be taken to distinguish between snowfall and snow accumulation because they are different. Also snow cover is a bit ambiguous and is typically used to refer to snow at a point rather than across a landscape, except at the end of the report then it's used to mean the latter. I think the use of snow cover can be avoided except for meaning across a landscape.

3. Often the authors refer to snowfall volume, yet they use one dimensional units of cm. Normal practice is to refer to precipitation amounts as depth in mm or cm. I suppose the authors could refer to specific volume, which also has units of mm or cm, but why complicate things. Unless they want to calculate volume of snow in a watershed, I'd stick with depth.

4. One important issue that is glossed over is uncertainty. If the data are compared then uncertainty needs to be included or the comparison has no context. Assessment of uncertainty, of both snowfall depth and duration, should be addressed in a separate paragraph.

5. Finally, I have a problem with the notion that one station can predict the snowfall at another station the following season. Do the authors think there is a teleconnection extending the 10-20 km between stations? By what physical process explains this phenomenon. Why aren't other station pairs predictors? How well are the stations correlated? How are we to know that this predictor is not a spurious correlation?

[M, D, & M] We are very grateful to Reviewer 2 for their extensive comments. The care
taken to point out the errors is immensely helpful to the success and impact of future publications. In revising this manuscript, we will give more attention to the language used to refer to precipitation versus snowfall, snow cover, and snow volume and call it out to make it more clear to the readers. We should have used snow depth rather than volume. Better care will be given to uncertainty. We incorrectly reported the uncertainty as error and that will be fixed. A section will be added to the discussion regarding uncertainty. The stations as predictors are discussed later in the reviewer responses. Another reviewer suggested a power analysis from Von Storch and Zwiers (2001) to analyze that relationship in more detail.

P1 L4: This is only part of the reason. The other part is the strong rain shadow exerted by the Transantarctic Mountains. See, Monaghan AJ, Bromwich DH, Powers JG and Manning KW (2005) The Climate of the McMurdo, Antarctica, Region as Represented by One Year of Forecasts from the Antarctic Mesoscale Prediction System*. Journal of Climate 18(8), 1174–1189 and see, Fountain et al., 2010

[M, D, & M] We propose updating this section to reflect that the MDVs lie within a rain shadow: "The Transantarctic Mountains buffer TV from the East Antarctic Ice Sheet (Chinn, 1990) and exert a rain shadow on the MDVs (Monaghan et al., 2005), thereby allowing them to remain ice-free and with little snow accumulation. Ephemeral streams supply melt from surrounding glaciers to . . ."

P2 L22: This reference is for sea ice, a more local reference, highlighting the effects on runoff from glaciers in the Dry Valleys, including the energy balance causes, is needed.

[M, D, & M] We agree, and the reference will be changed to Fountain et al., 1999 P3 L7: A more apt citation here, based on the physics of energy balance is Hoffman et al., 2016, already included in the reference list.

[M, D, & M] We agree. The citation will be changed from "Gooseff et al., 2011; Harpold and Brooks, 2018".

[Figure]

P3 L29: "They excluded..." It is not clear whether this study also excluded windy events or not.

[M, D, & M] Fountain et al. (2010) include accumulation during wind speeds > 5 m s-1 were as 'wind drift' and accumulation under lower wind speeds are termed 'direct precipitation'. This comment highlights one of the previously mentioned issues with this paper regarding the language used to discuss snowfall, snow cover, precipitation, etc. Because of the filtering above 5 m s-1, this paper only includes direct precipitation. The language will be changed to reflect more clearly when we discuss precipitation (i.e., rain and snow measured by the weighted gauge) and direct snowfall (i.e., accumulation measured by the ultrasonic sensors when wind speeds are less than 5 m s-1).

P4 L1: Abrupt and confusing transition from automatic measurements to a brief study of snow? It is unclear what you are trying to measure, not density, that is assumed to be 83 kg/m3, but then density was measured. Please clarify.

[M, D, & M] We are trying to measure density. To make this clearer, the paragraph will be updated beginning on P3 L29: "...0.5 mm water equivalent (w.e.). They converted ultrasonic distance ranger measurements of snow depth to w.e. using episodic measurements of density. A lack of published snow density records and logistical constraints limited snow density measurements to December of 2018 where we recorded a density of 83 kg m-3. Fountain et al. (2010) excluded precipitation events measured when daily average wind speeds exceed 5 m s-1 which could convey snow from the surrounding peaks to the valley floor. More details on station set up and data processing are described in detail by Fountain et al. (2010). Data are accessible from the MCM LTER website (http://mcm.lternet.edu/)." The MCM LTER reports snow densities measured on the glaciers, including the dates and thickness of the layers measured. These data will be analyzed and discussed to understand how a density of 83 kg m-3 impacts the results and perceived performance of the ultrasonic distance sensor.

P4 L9: "with Winter excluded for the same reason" Same as what? "Spring begins with

first light..." When is that?

[M, D, & M] We agree that this is vague and will update the sentence to: "Our study focuses on Spring through Fall, coincident with first and last light (September 1 and April 30; Acosta et al., 2020). This puts the seasonal and interannual variability of direct snowfall in the context of primary productivity and melt generation which are governed by available solar radiation. Summer falls from November 15 through February 15. Dates coincide with statistically distinct air temperature and solar radiation conditions (Obryk et al., 2020)."

P4 L10: "ends with final sunset" When is that? "Dates coincide with statistically distinct climate conditions" Climate is a very broad umbrella, which conditions, specifically?

[M, D, & M] It is based on the establishment of a statistically significant air temperature gradient with distance from the coast which is established by solar radiation. The atmosphere shifts from stable to unstable during the winter to summer transition as the increase in solar radiation warms the soil. Solar radiation also generates the onshore breeze which aids in establishing the gradient of increasing temperature with distance from the coast after temperature has been normalized to elevation. A more detailed discussion of this phenomena will be included in the discussion section 4.3 Implications for Hydrology and Ecology.

P4 L21: 'Instrumental' rather than "meteorological"?

[M, D, &M] Yes, thank you for pointing that out. This will be updated in the revised manuscript.

P4 L29: Why Commonwealth Glacier rather than Canada Glacier, which is adjacent to Lake Hoare? And why stake 23?

[M, D, & M] Commonwealth was chosen because it has stakes in the accumulation zone of the glacier. We regressed summer snow cover against all of the accumulation stakes on Commonwealth glacier and the strongest correlation was with stake 23,

which is why it was chosen. We will include a line about this in the manuscript.

P5 L5: "Precipitation" Is this a better descriptor or is snowfall? There is no rain and all precipitation is snow, so should the general term be used? I think not, it has a vagueness to it that is unnecessary.

[M, D, & M] It should be 'direct precipitation'. It includes measurements made by the ultrasonic and weighted gauge and the weighted gauge does not differentiate between rain and snow. Rain has been reported in the DVs, as recently as two years ago. Although it is very rare, we would rather not exclude it as a possibility in the weighted gauge measurements. This will be reflected in the manuscript. The usage of the terms direct precipitation, accumulation, snowfall, etc. will be described in better detail in the introduction and this section header will be updated to reflect those changes.

P5 L7: "Again"? When was it focused previously? Also considering that the seasons are partly defined by first and last light, if I understand the methods correctly, why is a 'light season' necessary? It's confusing. Also, this sentence underscores the confusing issue between precipitation and snowfall. Here you, 'focus on snow...', but the subheading is about precipitation, so the inference is that rain is being ignored.

[M, D, & M] These are all very good points and we struggled for a while with how to make it less confusing. We do include annual snowfall to compare our results with Fountain et al. (2010), but perhaps it would be best to discuss it in terms of snowfall and only specify when the snow is associated with a particular season. We agree with the confusion between snow and precipitation and will update the entire manuscript to highlight the differences in the terminology and use them properly because 'precipitation' and 'snow' are currently used interchangeably. Perhaps we could assume everything is snow to avoid confusion because rain is very rare.

P5 L13: 2007? Does this coincide with the results of Obryk on temporal break points?

[M, D, & M] Yes, Obryk sees a shift in temperature from increasing to no trend in

2007 at Lake Hoare and 2005 +/- 1 year at the other stations. The changepoint in the direct snowfall (because we are measuring with the ultrasonic) record is not statistically significant, but the trends on either side of it are. We don't discuss this very much, and a larger discussion section relating temperature to snowfall would improve the manuscript, particularly because the results of Obryk were a key motivator for this project.

P5 L18: This sentence doesn't seem logically connected. How is seasonal variability connected with differences in atmospheric influences? What influences are being considered?

[M, D, & M] The strength of Antarctic teleconnections is dependent on the season. By separating the analysis by season we may be able to isolate longer trends specific to and independent of these signals.

P5 L20: "spatial control" or spatial difference? Controls at this point in Results are unknown, but differences are known.

[M, D, & M] Yes, thank you for pointing that out. 'Spatial difference' is what is meant here and will be reflected in the manuscript.

P5 L21: You don't really mean 'volumes' right? Then it begs the question over what area are you measuring the volume. For precipitation, depth is the normal dimension used. Furthermore, the units of mm w.e. are not volume but a linear distance, so the dimensions of volume are wrong. Also, are these values averages? Please clarify. If yes, what is the standard deviation?

[M, D, & M] We mean depth here. These values are averages. The standard deviation will be reported in Table 1 with the seasonal means and the standard error of the mean.

P5 L23: If a third of w.e. snow occurs in spring and another third occurs in autumn (two thirds, not 'totaling over half', then one-third occurs in summer. If this is right, it doesn't square with the measurements at either the coastal or inland stations. Or am I

confused?

[M, D, & M] This sentence is confusing, but we agree that it is 'two thirds' rather than 'over half' and the sentence will be updated to: "Consequently, two thirds of the total August through May precipitation in TV occurs in the Lake Fryxell basin (FRLM and EXEM) during the Spring and Fall." We will also update Table 1 to include season totals, station totals, and a Taylor Valley total to make this more apparent.

P5 L25: "Bias". This brings up a couple of good points. First, do the authors mean 'bias' as in the measurements tend to be too high or too low? Second, do the authors mean 'uncertainty'. Considering they are comparing values, to make the comparison meaningful, they need to report uncertainty.

[M, D, & M] 'Bias' is not used properly here. We should have used 'uncertainty'. The uncertainties for each season, year, and station are reported in the supplemental file. The table, however, reports it as 'error' when it should be reported as 'uncertainty'. A careful examination of how we use 'error', 'uncertainty', and 'bias' will benefit the manuscript. Table 1 will be updated to show the uncertainty.

P5 L27: Looks like HOEM has a consistently lower seasonality than BOYM. Or is my interpretation due to missing data?

[M, D, & M] You are correct. We will update the paragraph to point toward HOEM as having the weakest seasonality. " BOYM (Figure 4d) has consistently low seasonality. Individual, large (> 2 mm w.e.) snow events can govern the season-scale fraction of snowfall for that year like we see in 2007 where Summer precipitation (9.5 mm w.e.) is nearly double that of Spring (5.0 mm w.e.). Low average precipitation and the occurrence of large snow events is likely responsible for its large interannual variability. We do not see any season-specific trend in snowfall at BOYM. HOEM has the weakest seasonality of all stations (Table 1; Figure 4c) and does not show any trends in snowfall for any season although data availability is limited to the last decade."

P5 L28: Revise, "where Summer precipitation (9.5 mm w.e.) is nearly" to "when Summer...was nearly".

[M, D, & M] This will be revised. The manuscript will be checked for this error throughout.

P5 L29: Again, another example of precipitation vs snowfall, "Low average precipitation and the occurrence of large snow events". So little rain and large snowfall?

[M, D, & M] This sentence is confusing and will be changed to: "Low average snowfall means that chance heavy snowfall events contribute to the interannual variability observed in the dataset."

P6 L1: Looks to me that EXEM has the greatest variability, not FRLM. Is that due to missing data at FRLM? It looks like prior to 2003 FRLM received no snowfall.

[M, D, & M] You are correct, EXEM does have the greatest variability. The supplemental table shows no data versus seasons with 0 mm w.e. accumulation. At FRLM, there was no data prior to 2002.

P6 L6: "stations are not predictors". I assume the authors do not mean the stations are correlated because in the following sentence Spring snowfall at FRLM predicts high summer snowfall at BOYM. I would have thought that a correlation matrix between stations would be included or perhaps referred to in supplementary data to support the notion that the station snowfall is not correlated between stations (is this right?). There is no physical reason for one station to predict snowfall at a later date at another station, unless it does so at the same station. Given the stations are only a few km apart the prediction is not based on the movement of air mass systems or a teleconnection. It's just persistence in the system.

[M, D, & M] We do mean they are correlated. We will include a correlation matrix as a supplemental file. I would argue that the reason we see heavy spring snowfall at FRLM indicative of heavy summer snowfall at BOYM due to the expansion of the

coastal climate further inland. That being said, we cannot explain why this relationship is only observed for FRLM and BOYM and not the other stations. This will be given more attention.

P6 L12: "snow cover heatmap"? Awkward, revise.

[M, D, & M] This sentence will be revised to: "Figure 5 suggests..."

P6 L14: "may inaccurately portray low snow cover for those seasons". It can't portray low snow cover, because like you say, the data is missing.

[M, D, & M] We intended for it to imply that there is some data missing for those years which may make the number of days with snow cover seem lower than it actually is. This will be revised beginning P6 L13: "...Spring and Fall, however a few days of data are missing during Spring and Fall of 2006, 2010, 2011, 2016, and 2017 snow years. The increase..."

P6 L15: "more gradual" than what? No rapid increase was identified.

[M, D, & M] This sentence will be removed.

P6 L17: replace 'high' with 'long'

[M, D, & M] Thank you for pointing that out. We will check for this error throughout the manuscript.

P6 L23: "snow cover at Lake Hoare is highly variable', clarify, snow cover persistence?

[M, D, & M] As was pointed out, snow cover is typically used to refer to an area rather than a point location. This distinction will be made when describing the language used regarding precipitation, snowfall, etc. Here, the sentence will be updated to "the fraction of days with snow on the ground".

P6 L25: Delete the last two sentences in the paragraph, they don't say anything.

[M, D, & M] We agree. These sentences will be removed.

[Figure]

P7 L29: Replace 'ground' with 'soil' or 'rock and soil'

[M, D, & M] We agree and will use 'rock and soil.'

P8 L1: Delete the sentence about sub-ice ecosystems, it doesn't go anywhere and is a distraction from the subject of the paragraph snowfall versus accumulation. If the ecosystem issue is important, develop in a separate paragraph.

[M, D, & M] We agree. This sentence will be moved to the last paragraph on P8 L30 just after "...snow accumulation at the valley-floor."

P8 L4: Delete 'necessary'. I'm sure one could figure a work-around if needed.

[M, D, & M] We agree. This will be removed.

P8 L6: Replace 'dissipation' with 'ablation', replace 'from' with 'based on'

[M, D, & M] Both of these changes will be made.

P8 L10: Replace 'pick up' with 'detect'

[M, D, & M] We will change this.

P8 L13: The last two sentences sort of repeat the last sentence in the previous paragraph. Can the influence of high winds be more fully addressed in a single paragraph rather than partly in two paragraphs?

[M, D, & M] Yes, these 3 sentences will be moved to the final paragraph on page 8 where wind is discussed again and is a reason for coupling measurement techniques.

P8 L18: " from cooling to no trend" Awkward. Rather than a 'shift', how about "a changing trend from cooling to no trend"?

[M, D, & M] We agree. The sentence will be updated to: "This shift coincides with a changing trend from cooling to no trend at Lake Hoare (Obryk et al., 2020)." We will still start the sentence with the word 'shift' in order to align with the previous sentence where we mention a 'shift' in interannual variability.

P8 L19: Delete the rest of the paragraph starting with, 'While the reconstructed...' The ablation and melt in the ablation zone vis a vis Hoffman et al., 2016 has nothing to do with snow cover. The text awkwardly summarizes the model and at the end of the paragraph, the authors back into a suggested process, reduced snow cover. Hoffman et al., do not argue that the lack of snow causes increased melt, they argue increased sediment on the ice surface. So, I don't know the purpose of these sentences.

[M, D, & M] We agree and the rest of that paragraph will be removed beginning with "While the reconstructed..."

P8 L27: Delete this paragraph, it doesn't make much sense. It starts to make an argument for snow cover vs snowfall relevance to local ecology. But that subject is dropped, and the subject shifts to the importance of winds again. It ends with an unsubstantiated statement about the best approach to measure snowfall and snow cover. Strangely, the ecosystem argument ignores an important aspect of snowfall and snow cover, its spatial distribution. In any case this paragraph doesn't really say anything important to the paper.

[M, D, & M] This paragraph will be replaced with one focused specifically on wind. Snowfall versus snow cover will be discussed in section 4.3 Implications for Hydrology and Ecology. This paper is more focused on the temporal characteristics rather than spatial characteristics of snow in TV.

P9 L3: "lowest relative loss of snow-covered area (72%)" What does this mean? And what is 72%? Does this mean that the area of Lake Hoare lost 72% of its snow cover? On average? Or is 72% of the Lake Hoare area covered in snow?

[M, D, & M] 72% of the area covered by snow in October 2009 completely ablated by January 2010 in the Lake Hoare basin. This is perhaps a better way to phrase this sentence and it will be rewritten as: "...2009-10 snow year. They found that 72% of the area covered by snow in October 2009 completely ablated by January 2010 in the Lake Hoare basin. Ninety-three percent and 97% of the snow-covered area in the Lake
Fryxell and Lake Bonney basins respectively ablated during the same time period. Lake Hoare lies..."

P9 L4: "least radiation" nuclear radiation?

[M, D, & M] This will be changed to solar radiation.

P9 L6: "may buffer reduced persistence associated with climatic conditions" Vague. What associated climate conditions?

[M, D, & M] This should say: "...buffer reduced persistence associated with higher RH, lower temperatures, and lower wind speeds."

P9 L7: volume, not depth? And the last sentence is very confused. How does snow at Lake Hoare inform on snow at the coast? "snow likely plays the larger role..." in what? And makes its monitoring increasingly important? Why not important, why is it increasing in importance? I might argue that it is more important to monitor snow up valley where there is less moisture.

[M, D, & M] We should have used depth here. There is a spatial gradient in snow depth described by Fountain. Near the coast there is more snow, so more light reduction for subnivean primary producers. This also depends on the photosynthetic efficiency of the communities. It is increasingly important because persistence is increasing and should be monitored at multiple locations to understand why. I agree that it would be great to monitor persistence along the valley (Fryxell, Hoare, and Bonney) to understand how climatic controls on persistence vary across the landscape.

P9 L9: Delete section 4.2? It doesn't come to any substantial conclusion. Given that the region is a desert and one large snow event can change the season of maximum snow fall, clearly the statistics will be very noisy and regressions and teleconnections will be insignificant. If the authors feel that this section is important, reduce it to one small tight paragraph.

[M, D, & M] We will delete this paragraph.

Interactive
comment

P9 L23: Delete 'sea ice extent' In the previous paragraph it was shown to be irrelevant to snowfall.

[M, D, & M] We agree it should be deleted

P10 L3: Delete section 4.3. This section lightly argues for the relevance of snow to the hydrology and ecology of Taylor Valley. Unfortunately, it's not particularly insightful and the topics have been better covered by the authors in the introduction. Also, there are several conceptual mistakes in 4.3 summarized below.

[M, D, & M] The discussion will be rewritten to focus more specifically on the dataset rather than hypothesize about potential causes of correlation.

P10 L7: "high-humidity areas which will experience greater melt" These areas will melt? Surely you don't mean that. Do you mean snow in those areas? "Sublimation is the greatest contributor to ablation of snow" Not true, in most temperate regions of the world melt is the biggest factor with sublimation playing a very small role.

[M, D, & M] This is referring to the fact that in high-humidity areas the snow is more likely to melt rather than sublimate. We should have clarified that this is specific to Taylor Valley. On pp 669, Gooseff et al. (2011) says "Sublimation is the most significant process ablating snow on the valley floors."

P10 L8: "Under these assumptions, reduced snow volume and increased snow per-sistence will further reduce the soil moisture contribution of snow which could have mixed effects on subsurface ice and soil communities. While there would be less melt to recharge subsurface ice, the increased duration of snow cover could act as a buffer and slow ablation." These two sentences are wrong showing a misunderstanding of the heat and mass transfer of snow over soil, particularly a relatively warm snow over much colder permafrost at depth.

[M, D, & M] We did a very poor job with word choice here and we agree with a previous comment that other authors have done a better job relating snow to hydrology and

ecology. The section on Implications for Hydrology and Ecology will be removed.

P10 L29: The predictive capability of high spring snowfall at FRLM to indicate high summer snowfall at BOYM only a short distance away is odd. This appears to be a case of correlation without causation, and not examined carefully by the authors. Furthermore it is very odd that no other station pairs show this, which makes me think this is specious and not worthy of inclusion in the conclusions.

[M, D, & M] This relationship will be excluded from the manuscript.

P11 L8: This sentence is unsubstantiated by anything prior in the report and should be deleted.

[M, D, & M] We propose instead to revise this sentence to: "A continued increase in snow cover and persistence increase the albedo of Taylor Valley which slows glacial melt, thereby slowing the increase in hydrologic and ecologic connectivity predicted by the MCM LTER (Wlostowski et al., 2016)." Because Taylor Valley hovers around 0°C during the summer an increase in albedo following a snow event can temporarily stop glacial melt. We mentioned this earlier in the manuscript (P2 L21).

P11 L11: This paragraph should be deleted; it doesn't say anything substantial.

[M, D, & M] We agree and that will be removed.

Figure 1. I've always thought that no acronyms should be included in a figure without explanation, otherwise, the reader has to search the text for interpretation. I recommend AWS be spelled out too.

[M, D, & M] The acronyms and AWS will be spelled out.

Figure 2. This is a confusing figure. The caption says the monitored area is outlined in red, but I only see a red line, not a polygon, so no 'area' is outlined. After some inspection I realized it was at the bottom of the photo. It would help the reader if the photo was cropped to minimize much of the sky in order to emphasize the monitored

[Figure]

area. 4 'perennially melted moat'? Is it always melted? The moat is part of the lake? Unclear The last sentence is awkward, please revise for clarity

[M, D, & M] We will crop the image and add labels to the figure. The reference to the 'moat' will be removed because it isn't discussed elsewhere in the manuscript.

Figure 4 I think it is important here to show missing data. Otherwise the plot is misleading, no bar is interpreted as zero snowfall. For example, it appears HOEM had snow snow accumulation between 1994 and 2006. The bar graph to the right, are these averages? If so, what is the sample size of each? It bears on the statistical differences between seasons and between stations.

[M, D, & M] We made a version of the figure showing missing data, but it distracted from the actual data. Maybe a light gray shading would be a good way to indicate where data are missing.

Figure 6. In the legend the mean is indicated by X-bar. But X-bar would be a snow year. Persistence is on the y axis, Y-bar? Why is the resolution of the photo +/- 0.5 days? This was not explained in the Methods.

[M, D, & M] That is a good point. The figure will be updated to y-bar. The photos are daily, so the resolution is half of the measurement which would be 0.5 days.

Figure 7. (b) where is the scale for accumulation? The tick marks suggest a scale different from (a).

[M, D, & M] The scale for accumulation is indicated on the left side of (a).

REFERENCES:

Fountain, A. G., Lyons, W. B., Burkins, M. B., Dana, G. L., Doran, P. T., Lewis, K. J., McKnight, D.M., Moorhead, D.L., Parsons, A.N., Priscu, J.C. and Wall, D.H. (1999). Physical controls on the Taylor Valley ecosystem, Antarctica. Bioscience, 49(12), 961-971. Von Storch, H. and Zwiers, F.W. (2001). Statistical analysis in climate research.

Cambridge university press.

---

## Author Comment (AC3) · 5 Jan 2021

AUTHOR RESPONSES TO REVIEWER 3 COMMENTARY ON MYERS ET AL MANUSCRIPT

This paper aims at describing snowfall and snow cover data that has been sampled in the Taylor Valley, in Antarctica. The protocol and the stations used to get the measurements are correctly described, and acquiring such data in this remote area is clearly a huge effort. The paper is well presented. However, the investigations based on these data are superficial, because many conclusions are not based on solid investigations, and some of them appear speculative. I would recommend to either publish this dataset in a web interface/journal with a DOI reference, or to conduct more investigations to prepare a manuscript. In such a form, I would recommend to reject this article.

There are two major points for which the study is not appropriate to make a scientific article to my point of view:

1. The trend analysis is based on very short timeseries, a point that strongly limits the possibility to evidence any climatic trend in the area. To my point of view, however, it would be interesting to provide a study focusing on the interannual variability. Such a study would require considering more variable/processes than the snowfall rates and snow persistence that are observed by the authors. In particular, the conclusions suggesting climatic signals in this area could be based on temperature/wind/pressure data, using observations at the local scale and potentially reanalysis at the regional scale.

This would lead credence to the section devoted to the teleconnections between the polar and the tropical to middle latitude areas.

2. The other weakness of this study is related to the links between snowfall and sea ice that are mentioned by the author through all the article, whereas there is not any sea-ice data in the study. In addition, the authors claim that a sea-ice reduction is expected in this area, that would favour a precipitation increase in relation to more moisture in the atmosphere. Even if such precipitation increase is expected in Antarctica under climate change, the sea-ice did not show any clear trend over the last decades, and even a slight increase in the Ross sea

[M, D, & M] We are very grateful to Reviewer 3 for their detailed comments which will greatly improve the readability and quality of the manuscript. In its current format the manuscript reads more like a data paper. The reference to Von Storch and Zwiers (2001) will be particularly useful in future analyses of this dataset and others. The discussion section will be refocused to discuss the results rather than giving so much weight to hypothesizing about what might be causing the trends we see.

LIST OF COMMENTS:

P2 L9: Annual mean of air temperature observed on average in TV by Obryk et al. is -20°C, so 18.5°C seems to warm (maybe a – sign is missing?).

[M, D, & M] Yes, the temperature should be negative and will be reflected in the revised manuscript.

P2 L27: The sea-ice extent in the Southern Hemisphere has been increasing over the last decades in particular in the area of the Ross Sea (de Santis et al., 2017), so should we expect a decrease in snowfall? This should be considered in the introduction and all over the manuscript.

[M, D, & M] More care will be taken to describing the relationship between snowfall and persistence and sea ice extent. We recently compared sea ice extent to the persistence dataset and saw a correlation during the Fall. This isn't discussed and will be added to the paper near P6 L20. More attention will be given to the decreasing direct snowfall, increasing persistence, and increasing sea ice extent in the Ross Sea.

P3 L25-30: It is claimed that the observation of precipitation is considered only when the wind is not exceeding 5 m s-1. But is it realistic to consider that there is no local snowfall with stronger winds? When the snow is drifted away with the wind, this does not mean that there is no snowfall, isn't it? I would expect more explanations for the situations when snowfall occurs during windstorms.

[M, D, & M] We agree, it is unrealistic to assume no accumulation under higher winds. This comment highlights an issue mentioned by another reviewer regarding the language used to discuss snowfall, snowcover, precipitation, etc. Fountain et al. (2010) categories accumulation during wind speeds > 5 m s-1 as 'wind drift' and accumulation under lower wind speeds are termed 'direct precipitation'. They use 5 m s-1 as the cutoff based on the definition of katabatic/foehn winds described by Nylen et al. (2004) and conditions during these wind events preclude snowfall. Because we exclude ac-

cumulation above 5 m s-1, the snow that is reported is direct precipitation or snowfall depending on whether or not it was measured by the weighted gauge or ultrasonic sensor respectively. The language will be updated to reflect these changes.

P4 L2: Is the snow density systematically equal to 83 kg m-3? That sounds like a strong assumption.

[M, D, & M] It is not. The issue of snow density was brought up by the other reviewers as well. The paragraph will be updated beginning on P3 L29: "...0.5 mm water equivalent (w.e.). They converted ultrasonic distance ranger measurements of snow depth to w.e. using episodic measurements of density. A lack of published snow density records and logistical constraints limited snow density measurements to December of 2018 where we recorded a density of 83 kg m-3. Fountain et al. (2010) excluded precipitation events measured when daily average wind speeds exceed 5 m s-1 which could convey snow from the surrounding peaks to the valley floor. More details on station set up and data processing are described in detail by Fountain et al. (2010). Data are accessible from the MCM LTER website (http://mcm.lternet.edu/)." We feel that the accuracy of our results could be impacted by the snow density measurements and will reach out to others who may have measured it. We will also include an additional paragraph in the discussion regarding how variability in snow density impacts our results derived from the ultrasonic sensor.

P4 L9: "Winter excluded for the same reason" -> which reason? The sentence is not clear. You mean that you do not focus on the winter season because of the lack of sunlight, isn't it? What are the limitations related to this protocol?

[M, D, & M] We agree that this is vague and will update the sentence to: "Our study focuses on Spring through Fall, coincident with first and last light (September 1 and April 30; Acosta et al., 2020). This puts the seasonal and interannual variability of direct snowfall in the context of primary productivity and melt generation which are governed by available solar radiation. Summer falls from November 15 through February 15.

Dates coincide with statistically distinct climate conditions (Obryk et al., 2020)." Little is known about the physical and ecological processes which occur in winter, but they are gaining more interest in the scientific community. Winter snowfall contributes to the mass balance of the surrounding glaciers. The impact of winter snow on ecology is an ongoing study.

Results: The discussion focusing on the volume of precipitation variability appears speculative, in particular because of the shortness of the time series as well as because of the missing data. The potential links between the spring snowfall at FRLM and the summer snowfall at BOYM is far from being clear visually. Even if the correlation is significant, would it be possible that this happened by chance? I would suggest providing also a power analysis (e.g. Von Storch and Zwiers, 2001) to estimate whether such a significant correlation has been obtained "by chance". The trends computed over such short periods should be considered very carefully also.

[M, D, & M] We will include a correlation matrix as a supplemental file. I would argue that the reason we see heavy spring snowfall at FRLM indicative of heavy summer snowfall at BOYM due to the expansion of the coastal climate further inland. That being said, we cannot explain why this relationship is only observed for FRLM and BOYM and not the other stations. This will be given more attention and a power analysis would benefit our interpretation of the correlation.

P5: Even shown in Figure 1, the names of the stations presented in the results and in particular in Figure 3 should be fully explained/detailed (BOYM, EXEM, HOEM, etc...)

[M, D, & M] The abbreviations will be removed from the figure.

P5 L20: What does mean the "c." before 0.5 mm in this sentence?

[M, D, & M] It means roughly or about or circa. We used 'c.' rather than '∼'.

P6 L2: A reference to Figure 4f is given whereas there is no visible f) in Figure 4.

[M, D, & M] It should say Figure 4b. There was originally a Figure 4f, but the figure was

edited and this sentence was not updated by accident.

P7 L22: "precipitation in terms of a snow year" -> Does it mean that the winter period is also included in the annual value? Or is the winter period excluded for the two sets of observation?

[M, D, & M] Yes, winter is included here. Winter is only included when we compare our dataset to the one published by Fountain et al. (2010). We will make this more clear in the introduction and methods.

P8 L13: Again, it could be interesting to give an estimation of both the spatial and the temporal variability of the snow density, because the choice of a constant value of 83 kg m-3 seems arbitrary. Also, it would be interesting to estimate the uncertainty of snowfall rates that directly emanate from the density uncertainty.

Did you consider to measure drifting snow, like Amory et al. (2020)?

[M, D, & M] The MCM LTER has snow density measurements from the glaciers with dates and we will comb through those data to see if any measurements were taken immediately following snowfall from our record. There is a lack of publications about snow density in the MDVs. We did not consider measuring drifting snow, but based on that paper, it seems unlikely that the dataset we present captures drifting snow because accumulation during wind speeds > 5 m s-1 is removed.

P9 L11: It is claimed that there is no correlation between snow cover in TV and sea-ice extent, but there is neither any figure, nor any number to evidence this finding. This finding should be illustrated with numbers or should appear in a previous publication. Same remark can be done with the temperature observations.

[M, D, & M] We will include a correlation matrix illustrating both of these.

P9 L28: "the increasing persistence may be indicative of the changing climate" -> This sounds very speculative, because the timeseries are very short, with missing data, the snow accumulation changes have a strong spatial variability as evidenced by the

signals that differ from one site to another. Also there is no analysis in the article concerning the mid-latitude to polar teleconnections, and in a general way the article does not include any investigations related to snowfall/snow persistence and atmospheric variables, which led few credence to the hypothesis appearing in the discussion.

[M, D, & M] We agree that it is speculative. It would be better to reword this to specify that the Fall is the least influenced by atmospheric oscillations and trends during the Fall may be representative of a background signal rather than reflective of atmospheric oscillations.

Implication for Hydrology and Ecology: The discussion related to the hydrological consequences is not clear. A situation with reduced snow volume and increased snow persistence (L.8) is pointed out as a situation that would favour a decrease of soil moisture. That's true, but if I have understood this article, I do not see in the previous section such opposite trends for snow persistence and snow accumulation.

[M, D, & M] Figure 3 shows increasing snow depth and Figure 6 shows the increasing persistence, although the increase in persistence is most notable in the Fall. This section will be moved immediately following the discussion section on trends in precipitation to highlight a discussion of our observations prior to discussing them.

P10 L8: "Sublimation is the greatest contributor to ablation of snow" -> Could you mention where such finding is applicable, please?

[M, D, & M] This is referring to the fact that in high-humidity areas the snow is more likely to melt rather than sublimate. In dry regions, snow is more likely to sublimate. We should have clarified that this is specific to Taylor Valley.

P10 L25: "Our record shows a clear increase in snowfall..." -> Could you remind where and when did you observe this trend, please?

[M, D, & M] This is a typo and should say 2007 rather than 2009. This is evident in Figure 3.

P10: "Snowfall has been decreasing trough 2017 [from 2009...], which contradicts the expected increase in snowfall in polar regions under warming conditions" -> To my mind, this sentence is a too-simplified view of the climate change in polar area, because this observed decrease of snowfall occurred only over 8 years (2009-2017), a short period for which the internal variability of the climate system can lead to any change of precipitation because of its chaotic nature, even if it is superimposed with the long-term warming trend observed over the last decades. Similarly, the authors could write an opposite sentence when they describe the increasing trend of snowfall that they observed between 1995 and 2009.

[M, D, & M] It may be better to rewrite this saying that the record isn't quite long enough yet to reveal any long-term controls of sea ice on snowfall in the MDVs. On shorter time scales (5-8 years), sea ice does not appear to influence snowfall.

P10 L28: The sentence has a grammatical issue, with a capital letter after a comma, and maybe a missing verb somewhere, and a blank space located at the middle of the sentence?

[M, D, & M] This will be corrected. A comma was accidentally used instead of a period. The space will be removed as well.

P11 L2: Again, it is claimed that there is no link between the increasing precipitation and the reduced sea ice, but there is no number evidencing any sea ice retreat, and such number cannot be found in the citation Fountain et al. (2010). Also, an impact of the synoptic-scale atmospheric conditions is suggested, but without any corresponding result shown in the article.

[M, D, & M] Sea ice has been expanding over the past decade and although snowfall has been declining, they do not correlate even when we introduced a lag in the data. It may be best not to discuss possible controls on snowfall and snow cover in the MDVs.

Data availability: If the article is published, the data used in this study should be made

available on a web interface, with a doi reference.

[M, D, & M] The data are accessible on the MCM LTER website and references were provided on P3 L25.

Table 1: Is the uncertainty shown in Table 1 includes the uncertainties of snowfall related to wind impact of sensors?

[M, D, & M] No, the uncertainty does not include wind. It is strictly related to the accuracy of the sensors.

Figure 1: What do you think about extending the area shown in Figure 1? This would allow to evidence that the Taylor Valley is a valley surrounded by mountains/glaciers.

[M, D, & M] The figure will be expanded.

Figure 2: a) and c) are mentioned in the caption, but not b).

[M, D, & M] (b) will be mentioned here: "The snow event captured in (b) has a persistence..."

Figure 5: The temporal resolution of the heatmap should be specified in the caption (daily resolution?).

[M, D, & M] Daily is correct. We will add it to the caption.

Figure 7: It seems that the number of days are centred over an average value, because there are negative values. This should be detailed in the caption.

[M, D, & M] This just highlights that the relationship does not do a good job at predicting low snow cover years. We will include a better description of this in the manuscript and figure caption.

References: Amory, C.: Drifting-snow statistics from multiple-year autonomous measurements in Adélie Land, East Antarctica, The Cryosphere, 14, 1713–1725, https://doi.org/10.5194/tc-14-1713-2020, 2020.
Angela De Santis, Eder Maier, Rodrigo Gomez & Inti Gonzalez (2017): Antarctica, 1979–2016 sea ice extent: total versus regional trends, anomalies, and correlation with climatological variables, International Journal of Remote Sensing, DOI:10.1080/01431161.2017.1363440

Von Storch, H. and Zwiers, F.W., 2001. Statistical analysis in climate research. Cambridge university press.